# Certified Robustness via Dynamic Margin Maximization and Improved Lipschitz Regularization

**Mahyar Fazlyab**[*], **Taha Entesari**,[*] **Aniket Roy, Rama Chellappa**
Department of Electrical and Computer Engineering
Johns Hopkins University
{mahyarfazlyab, tentesa1, aroy28, rchella4}@jhu.edu

## Abstract

To improve the robustness of deep classifiers against adversarial perturbations, many approaches have been proposed, such as designing new architectures with better robustness properties (e.g., Lipschitz-capped networks), or modifying the training process itself (e.g., min-max optimization, constrained learning, or regularization). These approaches, however, might not be effective at increasing the margin in the input (feature) space. In this paper, we propose a differentiable regularizer that is a lower bound on the distance of the data points to the classification boundary. The proposed regularizer requires knowledge of the model's Lipschitz constant along certain directions. To this end, we develop a scalable method for calculating guaranteed differentiable upper bounds on the Lipschitz constant of neural networks accurately and efficiently. The relative accuracy of the bounds prevents excessive regularization and allows for more direct manipulation of the decision boundary. Furthermore, our Lipschitz bounding algorithm exploits the monotonicity and Lipschitz continuity of the activation layers, and the resulting bounds can be used to design new layers with controllable bounds on their Lipschitz constant. Experiments on the MNIST, CIFAR-10, and Tiny-ImageNet data sets verify that our proposed algorithm obtains competitively improved results compared to the state-of-the-art.

## 1 Introduction

Motivated by the vulnerability of deep neural networks to adversarial attacks [1], i.e., imperceptible perturbations that can drastically change a model's prediction, researchers and practitioners have proposed various approaches to enhance the robustness of deep neural networks, including adversarial training [2–4], regularization [5, 6], constrained learning [7], randomized smoothing [8–10], relaxation-based defenses [11, 12], and model ensembles [13, 14]. These approaches modify the model architecture, the optimization procedure (loss function and algorithm), the inference mechanism, or the dataset itself [15] to enhance accuracy on both natural and adversarial examples.

However, most of these methods typically do not directly operate on input margins and rather target the output margin or its surrogates. For example, it is an established property that deep models trained with the cross-entropy loss, which is a surrogate for maximizing the output margin (see Appendix A.2), are prone to adversarial attacks [1]. From this perspective, it is critical to design regularized loss functions that can explicitly and effectively target the margin in the *input* space [16–18].

---

[*]Equal contribution.

37th Conference on Neural Information Processing Systems (NeurIPS 2023).

**Our Contribution** (1) Using first principles, we design a novel regularized loss function for training adversarially robust deep classifiers. We design a differentiable regularizer, using the Lipschitz constants of the logit differences, that is a lower bound on the input margin. We empirically show that this regularizer promotes larger margins in the input space. (2) We develop a scalable method for calculating guaranteed analytic and differentiable upper bounds on the Lipschitz constant of deep networks accurately and efficiently. Our method, called LipLT, hinges on the idea of *Loop Transformation* on the nonlinearities, which allows us to exploit the monotonicity and Lipschitz continuity of activations functions effectively. We prove that the resulting upper bound is better than the product of the Lipschitz constants of all layers (the so-called naive bound), and in practice, it is significantly better. Furthermore, our Lipschitz bounding algorithm can be used to design new layers with controllable bound on their Lipschitz constant. (3) We integrate our Lipschitz estimation algorithm within the proposed training loss to develop a robust training algorithm that eliminates the need for any inner-loop optimization subroutine. We utilize the recurring structure of the calculations that enable parallelized implementation on GPUs. When integrated into training, the relative accuracy of the bounds prevents excessive regularization and allows for more direct manipulation of the decision boundary.

Experiments on the MNIST, CIFAR-10, and Tiny-ImageNet data sets verify that our proposed algorithm obtains competitively improved results compared to the state-of-the-art. Code available on https://github.com/o4lc/CRM-LipLT.

## 1.1 Related Work

In the interest of space, we only review the most relevant literature and defer the comprehensive version to the supplementary materials.

**Enhancing Robustness via Optimization** The most well-known approach in adversarial defenses is perhaps adversarial training (AT) [4] and its certified variants [11, 19–23], which involves minimizing a worst-case loss (or its approximation) using uniformly-bounded perturbations to the training data. Since these approaches might hurt accuracy on non-adversarial examples, several regularization-based methods have been proposed to trade off adversarial robustness against natural accuracy [5, 6, 24, 25]. Intuitively, these approaches aim to control the variations of the model close to the decision boundary or around the training data points. In general, the main challenge is to formulate computationally efficient differentiable regularizers that can directly increase the margin of the classifier. Of notable recent work in this direction is [18], in which the authors design a regularizer that directly increases the margin in the input space and prioritizes more vulnerable points by leveraging the dynamics of the decision boundary during training. Our loss function also takes advantage of this approach, but rather than using an iterative algorithm to compute the margin, we use Lipschitz continuity arguments to find closed-form differentiable lower bounds on the margin.

**Robustness and Lipschitz Regularity** To obtain efficient and scalable certificates of robustness one can bound or control the global Lipschitz constant of the model. To train robust networks, the global Lipschitz bound is typically computed as the product of the spectral norm of linear layers (the naive bound), which provides computationally efficient but overly conservative certificates [5, 19, 26]. If these bounds are localized (e.g., in a neighborhood of each training data point), conservatism can be mitigated at the expense of losing computational efficiency [20, 27]. However, it is not clear if this approach provides any advantages over other local bounding schemes [28, 29] that can be more accurate with comparable complexity. Hence, for training purposes, it is highly desirable to have global but less conservative differentiable Lipschitz bounds. Of notable work in this direction is the semidefinite programming (SDP) approach of [30] (as well as its local version [31]) to provide accurate numerical bounds, which has also been leveraged in training low-Lipschitz networks [32]. However, these SDP-based approaches are restricted to small-scale models. In comparison, LipLT is less accurate than LipSDP but can scale to significantly larger models and larger input dimensions. Compared to the naive method, LipLT is significantly more accurate but has a comparable practical complexity thanks to a specialized GPU implementation that exploits the recursive nature of the algorithm.

**Lipschitz Layers** Another approach to improving robustness is to design new layers with controllable Lipschitz constant. Various methods have been proposed to design the so-called 1-Lipschitz networks [7, 33–37]. Inspired by the LipSDP framework of [30], the recent work [38] designs 1-Lipschitz layers that generalize many of the aforementioned 1-Lipschitz designs. However, their proposed approach is currently not applicable to multi-layer networks. In [39], the authors propose a reparameterization that directly satisfies the SDP constraint of LipSDP, but the proposed method cannot handle skip connections. [21] introduces $\ell_\infty$ distance neurons that are 1-Lipschitz and provide inherent robustness against $\ell_\infty$ perturbations.

## 1.2 Preliminaries and Notation

The log-sum-exp function is defined as $LSE(x) = \log(\sum_{i=1}^n \exp(x_i))$. For any $t > 0$, the perspective [40] of the log-sum-exp function satisfies the inequalities $\max_i(x_i) < t^{-1}LSE(tx) \leq \max_i(x_i) + t^{-1}\log(n)$. We denote the cross-entropy loss by $CE(q, p) = -\sum_{k=1}^K p_k \log(q_k)$, where $p, q \in \mathbb{R}^K$ are on the probability simplex. For an integer $y \in \{1, \cdots, K\}$, we define $u_y \in \mathbb{R}^K$ as the $y$-th standard unit vector. For a vector $a \in \mathbb{R}^K$, we have $\|a\|_p = (\sum_{i=1}^K |a_i|^p)^{1/p}$. Additionally, we have $\|a^\top\|_p = \|a\|_q$, where $1/p + 1/q = 1$. When $p = 1$, it is understood that $q = \infty$. For a matrix $A \in \mathbb{R}^{m \times n}$, $\|A\|$ denotes a matrix norm of $A$. In this paper we focus on $\|A\|_2 = \sigma_{\max}(A)$, i.e., the spectral norm of the matrix $A$. We, however, note that except for the implementation notes regarding the power iteration, the rest of the Lipschitz estimation algorithm is valid in any other matrix norm as well. We denote the indicator function of the event $E$ as $1_E$, where $1_E$ outputs 1 if the event $E$ is realized and 0 otherwise.

## 2 Certified Radius Maximization (CRM)

Consider a $K$-class deep classifier $C(x; \theta) = \arg\max_{1 \leq i \leq K} f_i(x; \theta)$, where $f(x; \theta) = \text{softmax}(z(x; \theta))$ is the vector of class probabilities, and $z(x; \theta)$ is a deep network. If a data point $x$ is classified correctly as $y \in \{1, \cdots K\}$, then the *logit margin* $\gamma(z(x; \theta), y)$, the difference between the largest and second-largest logit, is $\gamma(z(x; \theta), y) = z_y(x; \theta) - \max_{j \neq y} z_j(x; \theta)$.

Several existing loss functions for training classifiers are differentiable surrogates for maximizing the logit margin, including the cross entropy, and its combination with label smoothing (see Appendix A.2 for more details). However, maximizing the logit margin, or its surrogates, does not necessarily increase the margin in the input space. This can happen, for example, when the model has rapid variations close to the decision boundary. Thus, we need a regularizer targeted for maximizing the input margin.

**Regularization Based on Certified Radius** For a correctly-classified data $x$ with label $y$, the distance of $x$ to the decision boundary can be calculated by solving the optimization problem [18],

$$R(x, y; \theta) = \inf_{\hat{x}} \|x - \hat{x}\| \text{ subject to } \gamma(z(\hat{x}; \theta), y) = 0, \tag{1}$$

where the constraint enforces $\hat{x}$ to be on the decision boundary. [2] To ensure a large margin in the *input* space, we must, in principle, maximize the radius $R(x, y; \theta)$ of all correctly classified data points. To achieve this goal, we can add a regularizer that penalizes small input margins,

$$\min_\theta \mathbb{E}_{(x,y) \sim \mathcal{D}}[-\gamma(z(x; \theta), y)] + \lambda \mathbb{E}_{(x,y) \sim \mathcal{D}}[1_{\{\gamma(z(x;\theta),y)>0\}} g(R(x, y; \theta))], \tag{2}$$

where $\lambda > 0$ is the regularization constant and $g \colon \mathbb{R} \to \mathbb{R}$ is a decreasing function to promote larger certified radii. See appendix A.2 for more discussion on the properties of $g$.

To optimize (2), we must compute and differentiate through $R(x, y; \theta)$. For $\ell_\infty$ norm and ReLU activations, we can reformulate (1) as a mixed-integer linear program by using a binary representation

---

[2]We note that we can replace the equality constraint with the inequality constraint $\gamma(z(\hat{x}; \theta), y) \leq 0$ and get the same result.

of the activation functions. However, it is not affordable to solve this optimization problem during training. Hence, we must resort to computing lower bounds on $R(x, y; \theta)$ instead.

By replacing the logit margin with its soft lower bound in the constraint of (1), we obtain a lower bound on $R(x, y; \theta)$,

$$
\begin{aligned}
R(x, y; \theta) \geq \min_{\hat{x}} \quad & \|x - \hat{x}\| \\
\text{s.t.} \quad & z_y(x; \theta) - \frac{1}{t} \log \sum_{j \neq y} e^{t z_j(x; \theta)} \leq 0.
\end{aligned}
\tag{3}
$$

To solve this non-convex problem, the authors of [18] adapt the successive linearization algorithm of [41], which in turn does not provide a provable lower bound. To avoid this iterative algorithm and to provide sound lower bounds, we propose to use Lipschitz continuity arguments.

**Lipschitz-Based Surrogates for Certified Radius** Suppose we add a norm-bounded perturbation $\delta$ to a data point $x$ that is classified correctly as $y$. For the label of $x + \delta$ to not change, we must enforce

$$
\Delta z_{yi}(x + \delta) := z_y(x + \delta; \theta) - z_i(x + \delta; \theta) > 0 \quad \forall i \neq y.
$$

Suppose the logit difference $x \mapsto \Delta z_{yi}(x)$ is Lipschitz continuous with constant $L_{yi}$ (in $p$ norm), implying $|\Delta z_{yi}(x + \delta; \theta) - \Delta z_{yi}(x)| \leq L_{yi} \|\delta\|_p$, $\forall x, \delta$. Then we can write

$$
\Delta z_{yi}(x + \delta; \theta) \geq \Delta z_{yi}(x; \theta) - L_{yi} \|\delta\|_p \quad \forall i \neq y.
$$

The right-hand side remaining positive for all $i \neq y$ is a sufficient condition for the correct classification of $x + \delta$ as $y$. This sufficient condition yields a lower bound on $R(x, y; \theta)$ as follows,

$$
\underline{R}(x, y; \theta) := \min_{i \neq y} \frac{z_y(x; \theta) - z_i(x; \theta)}{L_{yi}} \leq R(x, y; \theta).
\tag{4}
$$

This lower bound is the pointwise minimum of logit differences normalized by their Lipschitz constants. This lower bound is not differentiable everywhere, but similar to the logit margin, we propose a smooth underapproximation of the $\min$ operator using scaled LSE:

$$
\underline{R}_t^{soft}(x, y; \theta) := -\frac{1}{t} \log \Big( \sum_{i \neq y} \exp \big( -t \frac{z_y(x; \theta) - z_i(x; \theta)}{L_{yi}} \big) \Big).
\tag{5}
$$

In addition to differentiability, another advantage of using this soft lower bound is that as opposed to $\underline{R}(x, y; \theta)$, which involves only two logits, the soft lower bound $\underline{R}_t^{soft}(x, y; \theta)$ includes all the logits, and hence, makes more effective use of information.

**Proposition 1** We have the following relationship between $\underline{R}_t^{soft}(x, y; \theta)$ defined in (5) and $\underline{R}(x, y; \theta)$ defined in (4).

$$
\underline{R}_t^{soft}(x, y; \theta) \leq \underline{R}(x, y; \theta) \leq \underline{R}_t^{soft}(x, y; \theta) + \frac{\log(K - 1)}{t}.
$$

In summary, we use the following loss function to train our classifier,

$$
\min_{\theta} \mathbb{E}_{(x,y) \sim \mathcal{D}}[-\underline{\gamma}(z(x; \theta), y)] + \lambda \mathbb{E}_{(x,y) \sim \mathcal{D}}[1_{\{\gamma(z(x;\theta), y) > 0\}} g(\underline{R}_t^{soft}(x, y; \theta))].
\tag{6}
$$

The first and the second terms are differentiable surrogates to the negative logit margin, and the input margin, respectively. For example, as we show in Appendix A.2 the negative cross-entropy loss is a differentiable lower bound on the logit margin, i.e., we can choose $\underline{\gamma}(z(x; \theta), y) = -CE(z(x; \theta), u_y)$.

It now remains to estimate the $L_{yi}$'s (the Lipschitz constants of $x \mapsto \Delta z_{yi}(x)$) that are needed to compute the second term in the loss function. While any guaranteed upper bound on $L_{yi}$ would suffice to preserve the lower bound property in (4), a more accurate upper bound prevents excessive regularization and allows for more direct manipulation of the decision boundary. To achieve this goal, we propose a new method which we will discuss next.

# 3 Scalable Estimation of Lipschitz Constants via Loop Transformation (LipLT)

In this section, we propose a general-purpose algorithm for computing a differentiable upper bound on the Lipschitz constant of deep neural networks, which is also of independent interest. For simplicity in the exposition, we first consider single hidden layer neural networks of the form $h(x) = W^1\phi(W^0 x)$, where $W^1, W^0$ have compatible dimensions, and the bias terms are ignored without loss of generality. The activation layers $\phi$ are of the form $\phi(z) = (\varphi(z_1), \cdots, \varphi(z_{n_1}))$ $z \in \mathbb{R}^{n_1}$, where $\varphi \colon \mathbb{R} \to \mathbb{R}$ is the activation function, which we assume to be monotone and Lipschitz continuous, implying $\alpha \leq (\varphi(x) - \varphi(x'))/(x - x') \leq \beta$ $\forall x \neq x'$ for some $0 \leq \alpha \leq \beta < \infty$ [30, 42]. In [30], the authors propose an SDP for computing an upper bound on the global Lipschitz constant of multi-layer neural networks when $\ell_2$ norm is considered in both input and output domains. This result for the single hidden layer case is formally stated in the following theorem.

**Theorem 1 ([30])** Consider a single-layer neural network described by $h(x) = W^1\phi(W^0 x)$. Suppose $\phi(x) \colon \mathbb{R}^{n_1} \to \mathbb{R}^{n_1} = [\varphi(x_1) \cdots \varphi(x_{n_1})]$, where $\varphi$ is slope-restricted over $\mathbb{R}$ with parameters $0 \leq \alpha \leq \beta < \infty$. Suppose there exist $\rho > 0$ and diagonal $T \in \mathbb{S}_+^{n_1}$ such that the matrix inequality

$$M(\rho, T) := \begin{bmatrix} -2\alpha\beta W^{0\top}TW^0 - \rho I_{n_0} & (\alpha+\beta)W^{0\top}T \\ (\alpha+\beta)TW^0 & -2T + W^{1\top}W^1 \end{bmatrix} \preceq 0, \tag{7}$$

holds. Then $\|h(x) - h(y)\|_2 \leq \sqrt{\rho}\|x - y\|_2$ for all $x, y \in \mathbb{R}^{n_0}$.

The key advantage of this SDP formulation is that we can exploit several properties of the structure, namely monotonicity ($\alpha \geq 0$) and Lipschitz continuity ($\beta < \infty$) of the activation functions, as well as using the same activation function in the activation layer. However, solving this SDP and enforcing them during training can be challenging even for small-scale neural networks. The recent work [43] has exploited the chordal structure of the resulting SDP imposed by the sequential structure of the network to solve the SDP for larger instances. However, these approaches are still unable to scale to larger problems and are not suitable for training purposes.

To guide the search for analytic solutions to the linear matrix inequality (LMI) of Theorem 1, the authors in [38] consider the following residual structure,

$$h(x) = Hx + G\phi(Wx). \tag{8}$$

Then it can be shown that the LMI condition (7) generalizes to condition (9) (see Appendix A.3 for details).

$$M(\rho, T) := \begin{bmatrix} -2\alpha\beta W^{\top}TW + H^{\top}H - \rho I_{n_0} & (\alpha+\beta)W^{\top}T + H^{\top}G \\ (\alpha+\beta)TW + G^{\top}H & -2T + G^{\top}G \end{bmatrix} \preceq 0, \tag{9}$$

By choosing $\rho = 1$, $H = I$ and $G = -(\alpha + \beta)W^{\top}T$, then the LMI (9) simplifies to $(\alpha+\beta)^2 TWW^{\top}T \preceq 2T$ (all blocks in the LMI except for the lower diagonal block become zero). When we restrict $T$ to be positive definite, then the latter condition is equivalent to $(\alpha+\beta)^2 WW^{\top} \preceq 2T^{-1}$, which can be satisfied analytically using various choices of $T$ [38]. In summary, the function $h(x) = x - (\alpha+\beta)W^{\top}T\phi(Wx)$ is guaranteed to be 1-Lipschitz as long as $(\alpha+\beta)^2 WW^{\top} \preceq 2T^{-1}$.

A potential issue with the above parameterization (and 1-Lipschitz networks in general) is that, since the true Lipschitz constant of the layer can be less than one, the multi-layer concatenation can become overly contractive. One way to resolve this limitation is to modify the parameterization as follows.

**Proposition 2** Suppose $WW^{\top} \preceq \frac{2\rho}{(\alpha+\beta)^2}T^{-1}$ for some $\rho > 0$ and some diagonal positive definite $T$. Then the following function is $\sqrt{\rho}$-Lipschitz.

$$h(x) = \sqrt{\rho}x - \frac{\alpha+\beta}{\sqrt{\rho}}W^{\top}T\phi(Wx). \tag{10}$$

Now if we make a cascade connection of $L \geq 2$ layers of the form (10) (each having its own $\rho$), a naive bound on the Lipschitz constant of the corresponding deep network would be $\prod_{i=1}^{L} \rho_i^{1/2}$. However, this upper bound can still be very crude. We now propose an alternative approach to compute an analytic upper bound on the Lipschitz constant of (8). For the multi-layer case, we then show that our method can capture the coupling between different layers to improve the naive bound. For the sake of space, we defer all the proofs to the supplementary materials.

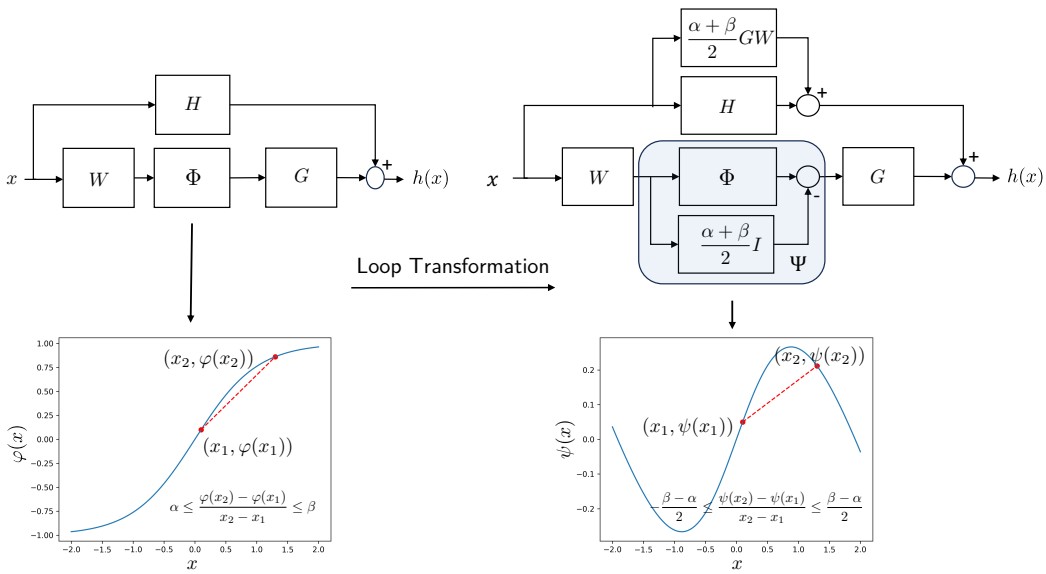

Figure 1: Loop transformation on a residual layer of the form $h(x) = Hx + G\phi(Wx)$. Here we use $\varphi(x) = \tanh x$ for the illustration of the loop transformation.

**The power of loop transformation** Starting from (8), using the fact that $\phi$ is $\beta$-Lipschitz, a global Lipschitz constant of $h$ can be computed as

$$\|h(x) - h(x')\| \le \|H(x - x')\| + \|G(\phi(Wx) - \phi(Wx'))\|$$
$$\le \underbrace{(\|H\| + \beta \|G\| \|W\|)}_{L_{\text{naive}}} \|x - x'\|. \tag{11}$$

This upper bound is pessimistic as it does not exploit the monotonicity ($\alpha \ge 0$) of activation functions. In other words, this bound would not change as long as $\alpha \ge -\beta$. To inform the bound with monotonicity, we perform a *loop transformation* [44, 45] to bring the non-linearities to the symmetric sector $[-(\beta - \alpha)/2, (\beta - \alpha)/2]$, resulting in the representation

$$h(x) = (H + \frac{\alpha + \beta}{2} GW)x + G\psi(Wx), \tag{12}$$

where $\psi(x) := \phi(x) - \frac{\alpha + \beta}{2} x$ is the loop transformed nonlinearity, which is no longer monotone, and is $\frac{\beta - \alpha}{2}$-Lipschitz– see Figure 1. We can now compute a Lipschitz constant as

$$\|h(x) - h(x')\| = \|(H + \frac{\alpha + \beta}{2} GW)(x - x') + G(\Psi(Wx) - \Psi(Wx'))\|$$
$$\le \|(H + \frac{\alpha + \beta}{2} GW)(x - x')\| + \|G(\Psi(Wx) - \Psi(Wx'))\|$$
$$\le \underbrace{(\|H + \frac{\alpha + \beta}{2} GW\| + \frac{\beta - \alpha}{2} \|G\| \|W\|)}_{L_{\text{LT}}} \|x - x'\|. \tag{13}$$

As we show below, this loop transformation does improve the naive bound. In Appendix A.3 we also prove the optimality of the transformation.

**Proposition 3** Consider a single-layer residual structure given by $h(x) = Hx + G\phi(Wx)$. Suppose $\phi \colon \mathbb{R}^{n_1} \to \mathbb{R}^{n_1}, \phi(x) = [\varphi(x_1) \cdots \varphi(x_{n_1})]$, where $\varphi$ is slope restricted in $[\alpha, \beta]$. Then $h$ is Lipschitz continuous with constant $\|H + \frac{\alpha + \beta}{2} GW\| + \frac{\beta - \alpha}{2} \|G\| \|W\| \le L_{\text{naive}}$.

**Bound refinement.** For the case of the $\ell_2$ norm, we can further improve the derived bound in (13). Specifically, we can bound the Lipschitz constant of the second term of $h$ in (12) analytically using LipSDP, which we summarize in the following theorem. [3].

**Theorem 2** Suppose there exists a diagonal $T$ that satisfies $G^\top G \preceq T$. Then a Lipschitz constant of (8) in $\ell_2$ norm is given by

$$L_{\text{LT-SDP}}(T) = \|H + \frac{\alpha + \beta}{2} GW\|_2 + \frac{\beta - \alpha}{2} \|W^\top TW\|_2^{\frac{1}{2}}.$$

By optimizing over the choice of diagonal $T$ satisfying $G^\top G \preceq T$, we can find the best Lipschitz bound achievable by this method. In this paper, however, we are more interested in analytical choices of $T$. Using the same techniques outlined in [38], we have the following choices:

1. Spectral Normalization (SN): We can satisfy $G^\top G \preceq T$ simply by choosing $T = T_{SN} := \|G\|_2^2 I$. In this case, $L_{\text{LT-SDP}}(T_{SN}) = \|H + \frac{\alpha+\beta}{2} GW\|_2 + \frac{\beta-\alpha}{2}\|W\|_2\|G\|_2 = L_{\text{LT}}$.

2. AOL [36]: Choose $T_{ii} = \sum_{j=1}^n |G^\top G|_{ij}$. As a result, $T - G^\top G$ is real-symmetric and diagonally dominant with positive diagonal elements, and thus, is positive semidefinite.

3. SLL [38]: Choose $T_{ii} = \sum_{j=1}^n |G^\top G|_{ij} \frac{q_j}{q_i}$, where $q_i > 0$. [38] uses the Gershgorin circle theorem to prove that this choice of $T$ satisfies $G^\top G \preceq T$.

## 3.1 Multi-layer Neural Networks

We now extend the proposed technique to the multi-layer case. Consider the following structure,

$$\begin{aligned}
y_k &= W_k x_k \quad k = 0, \cdots, L \\
x_{k+1} &= H_k x_k + G_k \phi(y_k) \quad k = 0, \cdots, L-1
\end{aligned} \tag{14}$$

with $W_L = I$ for consistency. Then $y_L = x_L$ is the output of the neural network. To compute a Lipschitz constant, we first apply loop transformation to each activation layer, resulting in the following representation.

**Lemma 1** Consider the sequences in (14). Define $\hat{H}_k := H_k + \frac{\alpha+\beta}{2} G_k W_k$ for $k = 0, \cdots, L-1$.[4] Then

$$y_{k+1} = W_{k+1}\hat{H}_k \cdots \hat{H}_0 x_0 + \sum_{j=0}^k W_{k+1}\hat{H}_k \cdots \hat{H}_{j+1} G_j \psi(y_j) \quad k = 0, \cdots, L-1. \tag{15}$$

We are now ready to state our result for the multi-layer case.

**Theorem 3** Let $m_k, k \geq 1$ be defined recursively as

$$m_{k+1} = \|W_{k+1}\hat{H}_k \cdots \hat{H}_0\| + \frac{\beta - \alpha}{2} \sum_{j=0}^k \|W_{k+1}\hat{H}_k \cdots \hat{H}_{j+1} G_j\| m_j \tag{16}$$

with $m_0 = \|W_0\|$. Then $m_k$ is a Lipschitz constant of $x_0 \mapsto y_k, k = 0, \cdots, L-1$. In particular, $m_L$ is a Lipschitz constant of the neural network $x_0 \mapsto y_L = x_L$.

Similar to the single-layer case, we can refine the bounds for the $\ell_2$ norm. For the sake of space, we defer this to the Appendix, and we use the bounds in (16) in our experiments.

**Complexity** For an $L$-layer network, the recursion in (16) requires $\mathcal{O}(L^2)$ number of matrix norm calculations. For the naive bound, this number is exactly $L$. Despite this increase in complexity, we utilize the recurring structure of the calculations that enable parallelized implementation on GPUs. This utilization results in a reduction of the time complexity to $\mathcal{O}(L)$, the same as the naive Lipschitz estimation algorithm. We provide a more detailed analysis of the computational and time complexity and the GPU implementation in Appendix A.4.

---

[3]The Theorem of [30] assumes that $\alpha \geq 0$ (monotonicity), but it can be shown that the theorem holds as long as $-\infty < \alpha$, which is the case for the loop-transformed nonlinearity.

[4]Note that we let $\hat{H}_k \cdots \hat{H}_{j+1}\big|_{j=k} = I$.

# 4 Experimental Results

In this section, we evaluate our proposed method for training deep classifiers on the MNIST [46], CIFAR-10 [47] and Tiny-Imagement [48] datasets. We compare our results with the closely related state-of-the-art methods. We further analyze the efficiency of our improved Lipschitz bounding algorithm and leverage it to compute the certified robust radii for the test dataset.

## 4.1 $\ell_2$-Robustness

**Experimental setup**   We train using our novel loss function to certify robustness (6), using cross-entropy as the differentiable surrogate, against $\ell_2$ perturbations of size $1.58$ on MNIST and $36/255$ on CIFAR-10 and Tiny-Imagenet [5]. We train convolutional neural networks of the form mCnF with ReLU activation functions, as in [20], where $m$ and $n$ denote the number of convolutional and fully connected layers, respectively. The details of the architectures, training process, and most hyperparameters are deferred to the supplementary materials.

**Baselines**   For baselines, we consider recent state-of-the-art methods that have provided results on convolutional neural networks of similar size. We consider: (1) GloRo [5] which uses naive Lipschitz estimation with a smoothness regularizer inspired by TRADES; (2) Local-lip-B/G (+ MaxMin) [20] which uses local Lipschitz constant calculation along with cut-off ReLU and MaxMin activation functions; (3) LipConvnet-5/10/15 [34] that uses 1-Lipschitz convolutional networks with GNP Householder activations; (4) SLL-small (SDP-based Lipschitz Layers) [38] which is a much larger 27 layer 1-Lipschitz network; (5) AOL-Small/Medium [36] which presents much larger 1-Lipschitz networks trained to have almost orthogonal layers.

**Evaluation**   We evaluate the accuracy of the trained networks under 3 criteria; standard accuracy, adversarial accuracy, and certified accuracy. For adversarial accuracy we perform PGD attacks using [4] (hyperparameters provided in supplementary material). For certified accuracy, we calculate the certified radii using (4) and verify if they are larger than the priori-assigned perturbation budget. For the methods of recent literature, we report their best numbers as per the manuscript.

**Results**   The results of the experiments are presented in Table 1. On MNIST, we outperform the state-of-the-art by a margin of 7.5% on verified accuracy whilst maintaining the same standard accuracy. On CIFAR-10 we surpass all of the state-of-the-art in verified accuracy. Furthermore, on methods using networks of similar sizes, i.e., 6C2F or LipConvnet-5, we surpass by a margin of around 4%. The networks of the recent literature AOL [36] and SLL [38] are achieving slightly worse certified accuracy even with much larger networks. On Tiny-Imagenet, our current best results are on par with the state-of-the-art.

### 4.1.1 Lipschitz Estimation

A key part of our method is the new improved Lipschitz estimation algorithm (LipLT) and the effective use of pairwise Lipschitz constants. Unlike previous works that estimate the pairwise Lipschitz between class $i$ and $j$ by the upper bound $\sqrt{2}L_z$ [20], where $L_z$ is the Lipschitz constant of the whole network, or as $L_i + L_j$ [5], where $L_i$ is the Lipschitz constant of the $i$-th class, we approximate this value directly as $L_{ij}$ by considering the map $z_i(x;\theta) - z_j(x;\theta)$. Table 2 shows the average statistic for Lipschitz constants estimated using different methods. Our improved Lipschitz calculation algorithm provides near an order-of-magnitude improvement over the naive method. Furthermore, Table 2 portrays the superiority of using pairwise Lipschitz constants instead of other proxies. Figure 2a illustrates the difference between using LipLT versus the naive method to calculate the certified radius. In this experiment, the naive method barely certifies one percent of the data points at the perturbation level of 1.58. However, using LipLT, we can certify a significant portion of

---

[5]We selected these values for the perturbation budget for consistency with prior work.

Table 1: Comparison with recent certified training algorithms. Best certified training accuracies are highlighted in bold.

* Due to a size mismatch that occurs in power iteration, we had to modify the architecture slightly by changing the padding of some of the convolutional layers. The number of neurons are the same as that of the original architecture in [20]. More details in the supplementary material

| Method | Model | Clean (%) | PGD (%) | Certified(%) |
|---|---|---|---|---|
| MNIST ($\epsilon = 1.58$) | | | | |
| Standard | 4C3F | 99.0 | 45.4 | 0.0 |
| GloRo | 4C3F | 92.9 | 68.9 | 50.1 |
| Local-Lip | 4C3F | 96.3 | 78.2 | 55.8 |
| CRM (ours) | 4C3F | 96.27 | 88.04 | **63.37** |
| CIFAR-10 ($\epsilon = 36/255$) | | | | |
| Standard | 6C2F | 87.5 | 32.5 | 0.0 |
| GloRo | 6C2F | 77.0 | 69.2 | 58.4 |
| Local-Lip-G | 6C2F | 76.4 | 69.2 | 51.3 |
| Local-Lip-B | 6C2F | 70.7 | 64.8 | 54.3 |
| Local-Lip-B + MaxMin | 6C2F | 77.4 | 70.4 | 60.7 |
| LipConvnet | 5-CR | 75.31 | - | 60.37 |
| LipConvnet | 10-CR | 76.23 | - | 62.57 |
| LipConvnet | 15-CR | 76.39 | - | 62.96 |
| SLL | Small | 71.2 | - | 62.6 |
| AOL | Small | 69.8 | - | 62.0 |
| AOL | Medium | 71.1 | - | 63.8 |
| CRM (ours) | 6C2F | 74.82 | 72.31 | **64.16** |
| Tiny-Imagenet ($\epsilon = 36/255$) | | | | |
| Standard | 8C2F | 35.9 | 19.4 | 0.0 |
| Local-Lip-G | 8C2F | 37.4 | 34.2 | 13.2 |
| Local-Lip-B | 8C2F | 30.8 | 28.4 | 20.7 |
| GloRo | 8C2F | 35.5 | 32.3 | 22.4 |
| Local-Lip-B + MaxMin | 8C2F | 36.9 | 33.3 | **23.4** |
| SLL | Small | 26.6 | - | 19.5 |
| CRM (ours) | 8C2F* | 23.97 | 23.04 | 17.98 |

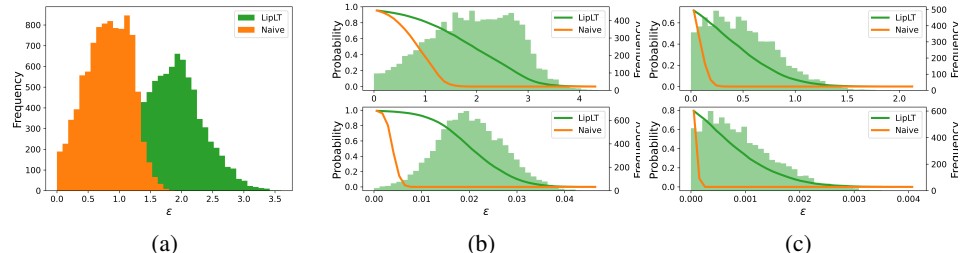

(a)     (b)     (c)

Figure 2: (a) Distribution of certified radii calculated using direct pairwise Lipschitz constants for the MNIST test dataset for the network trained using CRM. (b-c) Comparison of the distribution of the certified radii for a model trained using CRM (top) versus a standard trained model (bottom) for MNIST (b) and CIFAR-10 (c). For any given $\epsilon$, the probability curves denote the empirical probability of a data point from that data set having a certified radius of at least $\epsilon$.

the data points.

We further conduct an analysis of the certified radii of the data points for regularized versus unregularized networks for MNIST and CIFAR-10. Figures 2b and 2c illustrate the distribution of the certified radii. These radii were computed using the direct pairwise Lipschitz bounding. When comparing the results of the CRM-trained models (top figures) with the standard models (bottom figures), it becomes evident that our loss function enhances the certified radius.

Table 2: Comparison of average pairwise Lipschitz constant calculated using the naive and improved method on networks trained using CRM. We use the same architectures as Table 1. To calculate the averages, we form all unique pairs and calculate the Lipschitz constant according to the relevant formulation and then take the average.

| | MNIST | | | CIFAR10 | | | Tiny-Imagenet | | |
|---|---|---|---|---|---|---|---|---|---|
| | $L_{ij}$ | $L_i + L_j$ | $\sqrt{2}L$ | $L_{ij}$ | $L_i + L_j$ | $\sqrt{2}L$ | $L_{ij}$ | $L_i + L_j$ | $\sqrt{2}L$ |
| Naive | 266.08 | 1285.18 | 2759.63 | 93.52 | 131.98 | 139.75 | 313.62 | 485.62 | 2057.00 |
| Improved | 64.10 | 212.68 | 419.83 | 11.26 | 18.30 | 23.43 | 9.46 | 14.77 | 60.37 |

# 5    Limitations

Although our proposed formulation has an elegant mathematical structure, there exist some limitations: **(1)** We are still using global Lipschitz constants to bound the margin, which can be conservative. We will investigate how we can localize our calculations without a significant increase in computation. **(2)** Computation of pairwise Lipschitz bounds for very deep architectures and or a large number of classes can become computationally intensive for training purposes (see Appendix A.4 for further discussions). **(3)** Several hyper-parameters require manual tuning. It is highly desirable to explore adaptive approaches to automatically select these hyper-parameters.

# 6    Conclusion

Adversarial defense methods are numerous and their approaches are different, but they all attempt to explicitly or implicitly increase the margin of the classifier, which is a measure of adversarial robustness (or vulnerability). From this perspective, it is highly desirable to develop adversarial defenses that can manipulate the decision boundary and increase the margin effectively and efficiently. We attempt to maximize the input margin by penalizing the Lipschitz constant of the neural network along vulnerable directions. Additionally, we develop a new method for calculating guaranteed analytic and differentiable upper bounds on the Lipschitz constant of the deep network. LipLT is provably better than the naive Lipschitz constant. We have also provided a parallelized implementation of LipLT using the recurring structure of the calculations, which is fast and scalable. Our proposed method achieves competitive results in terms of verified accuracy on the MNIST and CIFAR-10 datasets.

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
