# A    Supplementary Material

## A.1    Additional Literature Review

Here we provide additional literature review of adversarial robustness.

**Estimation of Lipschitz Constant**    In the literature, there exist several approaches for estimating both the global and local Lipschitz constant of deep neural networks [49–54]. While local bounds can be obtained using bound propagation-like methods [20, 27, 49, 55], bounding the global Lipschitz constant appears to be a more challenging problem. Combettes et al. [56] use monotone operator theory to derive global analytic bounds. Fazlyab et al. [30] propose LipSDP, which computes numerical upper bounds on the global Lipschitz constant based using SDP.

**Margin Maximization**    The distance of data points to the decision boundary quantifies the margin of a classifier. The fact that maximizing margin provides robust models has been adopted in, for example, standard max-margin classifiers [57], and SVMs [58]. For deep neural networks, since computing the exact margin is difficult, some approximations are pursued as surrogates. For example, [59, 60] uses first-order Taylor approximation of margin. See [61] for a recent survey on large-margin learning.

**Adversarial Training**    Weighted adversarial training methods tend to provide higher weights to the more vulnerable points closer to the decision boundary. For example, MMA [16] applies cross-entropy loss on only the points closest to the decision boundary. GAIRAT [17] re-weights the adversarial samples based on the least number of iterations needed to flip the clean sample to an adversarial example and uses that as a surrogate to the margin. MAIL [62] reweights adversarial samples based on their logit margin.

**Certified Training**    Certified robustness methods provide guarantees of robustness by optimizing an upper bound on a worst-case loss. These upper bounds are typically obtained by bound propagation [63, 64], convex relaxations, or Lipschitz continuity arguments. For instance, [65, 66] use semidefinite relaxations instead to obtain differentiable surrogates for the robust cross-entropy loss. However, solving these SDPs during training is practically infeasible. More recently, Mao et al. [67] established a connection between certified and adversarial training (TAPS), yielding precise worst-case loss approximation, and thus reduced over-regularization and increased certified and standard accuracies.

We finally note that adversarial robustness can also be promoted during inference time. The most well-known approach is randomized smoothing [8], which provides a guaranteed lower bound on the certified radius.

## A.2    Supplementary Materials for Section 2

**Differentiable Surrogates for Logit Margin.**    Since the logit margin $\gamma$ is non-differentiable, optimizing it would likely slow down the convergence. Replacing the $\max$ operator with LSE, we obtain a differentiable approximation of the logit margin,

$$\gamma_t^{\text{soft}}(z(x;\theta),y) = z_y(x;\theta) - \frac{1}{t}\log\sum_{j\neq y} e^{tz_j(x;\theta)}, \tag{17}$$

where $t > 0$. Using the inequality satisfied by the perspective of LSE (see § 1.2), we obtain the following bounds,

$$\gamma_t^{\text{soft}}(z(x;\theta),y) \leq \gamma(z(x;\theta),y) \leq \gamma_t^{\text{soft}}(z(x;\theta),y) + \frac{\log(K-1)}{t}$$

In other words, $\gamma_t^{\text{soft}}(z(x;\theta),y)$ is a differentiable lower bound to the logit margin and the gap shrinks as $t$ increases. For $t=1$, we obtain

$$\gamma_1^{\text{soft}}(z(x;\theta),y) = z_y(x;\theta) - \log\sum_{j\neq y} e^{z_j(x;\theta)} \geq \log(e^{z_y(x;\theta)}) - \log\sum_j e^{z_j(x;\theta)} = -CE(f(x;\theta),u_y)$$

In other words, the negative cross-entropy loss is a lower bound on the soft version of the logit margin.

**Choice of Function $g$**   As elaborated in [18], the function $g$ must be decreasing to prioritize vulnerable data points, i.e., $g$ must assign a higher cost to points that are closer to the decision boundary. However, this regularization may cause a conflict among the data points: the decision boundary tends to move towards vulnerable points (points with small margins), which conflicts with the goal of robust training. [18] identifies strict convexity as an additional desired property for $g$ to mitigate this conflict. Despite this, we did not measure any significant improvement added by strict convexity of $g$. Consequently, we have used the linear function outlined in the experiments section. In future work, we will explore the possibility of learning the optimal $g$ during the training phase.

**Differentiable Lower Bound on Certified Radius**   We recall the definition of certified radius in (1). In this definition, we can replace equality with inequality without changing the optimal value,

$$R(x,y;\theta) = \{\inf_{\hat{x}} \|x - \hat{x}\| \mid \gamma(z(\hat{x};\theta),y) \leq 0\}, \tag{18}$$

To see this, suppose $x$ is a correctly classified point ($\gamma(z(x;\theta),y) > 0$), and $\hat{x}^\star$ is an optimal solution of (18) that satisfies $\gamma(z(\hat{x}^\star;\theta),y) < 0$. Define $\hat{x}(t) = \hat{x}^\star + t(x - \hat{x}^\star)$, where $t \in [0,1]$. Since $\gamma(z(\hat{x}(0);\theta),y) < 0$ and $\gamma(z(\hat{x}(1);\theta),y) > 0$, by continuity of $\gamma$, we can conclude that there exists $\hat{t} \in (0,1)$ such that $\gamma(z(\hat{x}(\hat{t});\theta),y) = 0$. Now we can write

$$\|x - \hat{x}(\hat{t})\| = \|(1-\hat{t})(x - \hat{x}^\star)\| < \|x - \hat{x}^\star\|.$$

In other words, the point $\hat{x}(\hat{t})$ is on the decision boundary and improves the cost function in (18). Thus, we can replace the equality constraint with the inequality constraint.

Finally, we note the soft logit margin defined in (17) is a lower bound on the logit margin itself. Thus, we can write

$$\{\hat{x} \mid \gamma(z(\hat{x};\theta),y) \leq 0\} \subseteq \{\hat{x} \mid \gamma_t^{soft}(z(\hat{x};\theta),y) \leq 0\}.$$

This implies the inequality in (3).

**Proof of Proposition 1**

**Proposition 1** We have the following relationship between $\underline{R}_t^{soft}(x,y;\theta)$ defined in (5) and $\underline{R}(x,y;\theta)$ defined in (4).

$$\underline{R}_t^{soft}(x,y;\theta) \leq \underline{R}(x,y;\theta) \leq \underline{R}_t^{soft}(x,y;\theta) + \frac{\log(K-1)}{t}.$$

**Proof** Starting from (4), we can write

$$\underline{R}(x,y;\theta) := -\max_{i\neq y} \frac{-(z_y(x;\theta) - z_i(x;\theta))}{L_{yi}} \leq R(x,y;\theta).$$

By approximating the maximum operator with the perspective of the log-sum-exp function, we obtain (5),

$$\underline{R}_t^{soft}(x,y;\theta) := -\frac{1}{t}\log\left(\sum_{i\neq y}\exp\left(-t\frac{z_y(x;\theta) - z_i(x;\theta)}{L_{yi}}\right)\right).$$

Finally, by using the inequalities satisfied by the perspective of LSE (see §1.2), we obtain the desired inequality (4).

**Regularizing the Lipschitz Constant of the Network**  We note that $z_y(x;\theta) - z_i(x;\theta) = (u_y - u_i)^\top z(x;\theta)$. We can then write

$$L_{yi} = \mathrm{Lip}_p((u_y - u_i)^\top z(x;\theta)) \le \|(u_y - u_i)^\top\|_p \mathrm{Lip}_p(z(x;\theta)) = \|u_y - u_i\|_q \mathrm{Lip}_p(z(x;\theta)),$$

where $1/p + 1/q = 1$ (recall that $u_i$ denotes the $i$-th unit vector), and $\mathrm{Lip}_p(.)$ denotes the Lipschitz constant in $\ell_p$ norm. Since $\|u_y - u_i\|_q = 2^{1/q}$ is independent of $i$, we can conclude

$$\underline{R}(x, y; \theta) := \min_{i \ne y} \frac{z_y(x;\theta) - z_i(x;\theta)}{L_{yi}} \ge \frac{2^{-1/q}}{\mathrm{Lip}_p(z(x;\theta))} \gamma(z(x;\theta), y). \tag{19}$$

In other words, we can find yet another lower bound on the input margin by using the Lipschitz constant of the *whole network* (the logit vector $z(x;\theta)$). Using this relatively crude lower bound, we can enlarge the input margin by maximizing the logit margin $\gamma(z(x;\theta), y)$ or its surrogates while minimizing the Lipschitz constant of the whole network,

$$\min_\theta \mathbb{E}_{(x,y)\sim\mathcal{D}}[-\gamma(z(x;\theta), y)] + \lambda \mathbb{E}_{(x,y)\sim\mathcal{D}}[\mathrm{Lip}_p(z(x;\theta))].$$

Therefore, regularizing the Lipschitz constant of the whole network has a less direct effect on the margin, according to the lower bound in (19).

## A.3   Supplementary Materials for Section 3

**Feasibility of LipSDP.**  Theorem 1 requires the existence of $\rho > 0$ and positive semi-definite $T$. Such $\rho$ and $T$ always exist. To see this, let $T = \frac{1}{2}\|W^1\|_2^2 I_{n_1}$, so that $-2T + W^{1\top}W^1 \preceq 0$ holds. Now, by applying the Schur complement we arrive at the equivalent condition

$$(\alpha + \beta)^2 W^{0\top} T(2T - W^{1\top}W^1)^\dagger TW^0 - 2\alpha\beta W^{0\top}TW^0 \preceq \rho I_{n_0}.$$

Any value of $\rho$ larger than the largest singular value of the matrix on the left-hand side is valid. As a result, the LMI is always feasible.

**Detailed Derivation of** (9)   As per the arguments of [30], for an activation function $\phi$ slope-bounded between $\alpha$ and $\beta$, and a positive semidefinite diagonal matrix $T$, we have

$$\begin{bmatrix} x - y \\ \phi(x) - \phi(y) \end{bmatrix}^\top \begin{bmatrix} -2\alpha\beta T & (\alpha + \beta)T \\ (\alpha + \beta)T & -2T \end{bmatrix} \begin{bmatrix} x - y \\ \phi(x) - \phi(y) \end{bmatrix} \ge 0, \quad \forall x, y \in \mathbb{R}^n.$$

We can similarly state this for when inputs are $Wx$ and $Wy$ instead:

$$\begin{bmatrix} Wx - Wy \\ \phi(Wx) - \phi(Wy) \end{bmatrix}^\top \begin{bmatrix} -2\alpha\beta T & (\alpha + \beta)T \\ (\alpha + \beta)T & -2T \end{bmatrix} \begin{bmatrix} Wx - Wy \\ \phi(Wx) - \phi(Wy) \end{bmatrix} \ge 0.$$

This is equivalent to

$$\underbrace{\begin{bmatrix} x - y \\ \phi(Wx) - \phi(Wy) \end{bmatrix}}_{z(x,y)}^\top \underbrace{\begin{bmatrix} -2\alpha\beta W^\top TW & (\alpha + \beta)W^\top T \\ (\alpha + \beta)TW & -2T \end{bmatrix}}_{A} \begin{bmatrix} x - y \\ \phi(Wx) - \phi(Wy) \end{bmatrix} \ge 0.$$

Now, suppose the operator $h(x) = Hx + G\phi(Wx)$ is $\sqrt{\rho}$ Lipschitz. Then for all $x$ and $y$ we have

$$\|Hx + G\phi(Wx) - (Hy + G\phi(Wy))\|^2 \le \rho\|x - y\|^2.$$

This can equivalently be stated in the following form

$$z(x, y)^\top \underbrace{\begin{bmatrix} H^\top H - \rho I & H^\top G \\ G^\top H & G^\top G \end{bmatrix}}_{B} z(x, y) \le 0.$$

Since $z(x, y)^\top A z(x, y) \ge 0$ for all $x, y$, by imposing $B + A \preceq 0$, we will have $z(x, y)^\top B z(x, y) \le 0$ for all $x, y$.

**Detailed Derivation of** (11)  If $f$ and $g$ are $L_f$- and $L_g$-Lipschitz, respectively, then $f \circ g$ and $f + g$ (assuming compatible dimensions), is $(L_f \cdot L_g)$-Lipschitz, and $(L_f + L_g)$-Lipschitz, respectively [19]. Using these properties:

1. $x \mapsto Hx$ is $\|H\|$-Lipschitz.

2. $x \mapsto \phi(x)$ is $\beta$-Lipschitz,

$$\|\phi(x) - \phi(x')\|_p = \left( \sum_{i=1}^{n} |\varphi(x_i) - \varphi(x'_i)|^p \right)^{1/p} \leq \beta \left( \sum_{i=1}^{n} |x_i - x'_i|^p \right)^{1/p} = \beta \|x - x'\|_p.$$

3. $x \mapsto (G \circ \phi \circ W)x$ is $(\beta \|G\| \|W\|)$-Lipschitz.

4. Finally, from 1. and 3., $x \mapsto Hx + G\phi(Wx)$ is $(\|H\| + \beta \|G\| \|W\|)$-Lipschitz.

**Proof of Proposition 2**

**Proposition 2** Suppose $WW^\top \preceq \frac{2\rho}{(\alpha+\beta)^2} T^{-1}$ for some $\rho > 0$ and some diagonal positive definite $T$. Then the following function is $\sqrt{\rho}$-Lipschitz.

$$h(x) = \sqrt{\rho} x - \frac{\alpha + \beta}{\sqrt{\rho}} W^\top T \phi(Wx). \tag{10}$$

**Proof** For a given $\rho > 0$, the LMI in (9) ensures that $h$ is $\sqrt{\rho}$-Lipschitz. By choosing $H = \sqrt{\rho} I$ and $G = -\frac{\alpha+\beta}{\sqrt{\rho}} W^\top T$, this LMI simplifies to

$$M(\rho, T) := \begin{bmatrix} -2\alpha\beta W^\top TW & 0 \\ 0 & -2T + \frac{(\alpha+\beta)^2}{\rho} TWW^\top T \end{bmatrix} \preceq 0. \tag{20}$$

Since $-2\alpha\beta W^\top TW \preceq 0$ holds for any $T \succ 0$, the LMI (20) is equivalent to $WW^\top \preceq \frac{2\rho}{(\alpha+\beta)^2} T^{-1}$.

**Remark** *The condition $T \succ 0$ can be relaxed to $T \succeq 0$. We noted that given the aforementioned choice of the parameters, the LMI (9) is equivalent to $TWW^\top T \preceq \frac{2\rho}{(\alpha+\beta)^2} T$. If we restrict the choice of $T$ to be zero on certain indices denoted by $J \subset \{1, \cdots, n_1\}$, then this condition is equivalent to $(WW^\top)_{J^C} \preceq \frac{2\rho}{(\alpha+\beta)^2} T_{J^C}^{-1}$, where $J^C$ is the complement of $J$, and for a matrix $A$, $A_{J^C}$ is the principal minor of $A$, i.e., the submatrix that remains after removing rows and columns with indices in $J$.*

**Proof of Proposition 3**

**Proposition 3** Consider a single-layer residual structure given by $h(x) = Hx + G\phi(Wx)$. Suppose $\phi : \mathbb{R}^{n_1} \to \mathbb{R}^{n_1}, \phi(x) = [\varphi(x_1) \cdots \varphi(x_{n_1})]$, where $\varphi$ is slope restricted in $[\alpha, \beta]$. Then $h$ is Lipschitz continuous with constant $\|H + \frac{\alpha+\beta}{2} GW\| + \frac{\beta-\alpha}{2} \|G\| \|W\| \leq L_{\text{naive}}$.

**Proof** Since $\varphi$ is slope-restricted in $[\alpha, \beta]$, by definition we can write (see [30, 42] for more context on slope-restricted activation functions),

$$\alpha \leq \frac{\varphi(x) - \varphi(x')}{x - x'} \leq \beta \quad \forall x \neq x'.$$

Subtracting $\frac{\alpha+\beta}{2}$ from both sides, we obtain $-\frac{\beta-\alpha}{2} \leq \frac{\varphi(x) - \varphi(x') - \frac{(\alpha+\beta)}{2}(x-x')}{x-x'} \leq \frac{\beta-\alpha}{2}$, or, equivalently, $|\varphi(x) - \varphi(x') - \frac{(\alpha+\beta)}{2}(x - x')| \leq \frac{\beta-\alpha}{2}(x - x')$. In words, the transformed nonlinearity $\varphi(x) - \frac{\alpha+\beta}{2} x$ is $(\beta - \alpha)/2$-Lipschitz but is no longer monotone.

Define $\psi(x) = \phi(x) - \frac{\alpha+\beta}{2}x$. By the preceding inequality, we can conclude that $\psi$ is Lipschitz with constant $(\beta - \alpha)/2$:

$$\|\phi(x) - \phi(x') - \frac{\alpha+\beta}{2}(x-x')\|_p = \left(\sum_{i=1}^{n_1} |\varphi(x_i) - \varphi(x_i') - \frac{\alpha+\beta}{2}(x_i - x_i')|^p\right)^{1/p}$$

$$\leq \left(\sum_{i=1}^{n_1}(\frac{\beta-\alpha}{2})^p|x_i - x_i'|^p\right)^{1/p}$$

$$= \frac{\beta-\alpha}{2}\|x - x'\|_p.$$

Using the definition of $\psi$, we can rewrite $h$ as

$$h(x) = (H + \frac{\alpha+\beta}{2}GW)x + G\psi(Wx).$$

We can now compute a Lipschitz constant for $h$ as follows,

$$\|h(x) - h(x')\| = \|(H + \frac{\alpha+\beta}{2}GW)(x-x') + G(\psi(Wx) - \psi(Wx'))\|$$

$$\leq \|(H + \frac{\alpha+\beta}{2}GW)(x-x')\| + \|G(\psi(Wx) - \psi(Wx'))\|$$

$$\leq \underbrace{(\|H + \frac{\alpha+\beta}{2}GW\| + \frac{\beta-\alpha}{2}\|G\|\|W\|)}_{L_{\text{LT}}}\|x - x'\|$$

Finally, we can write

$$L_{\text{LT}} \leq \|H\| + \frac{\alpha+\beta}{2}\|GW\| + \frac{\beta-\alpha}{2}\|G\|\|W\| \leq \|H\| + \beta\|G\|\|W\| = L_{naive},$$

where the first and second inequalities follow from the sub-additive and sub-multiplicative properties of matrix norms, respectively. The proof is complete.

**On the Optimality of Loop Transformation** In the proof of Proposition 3 (and its multi-layer counterpart Theorem 3), we shifted the nonlinearity by $\frac{(\alpha+\beta)}{2}x$ and showed that this indeed improves the naive bound. However, it is not clear whether this is optimal in the sense of obtaining the tightest upper bound on the Lipschitz constant. To study this formally, we consider the shifted nonlinearity $\psi(x) = \phi(x) - \gamma x$, where $\gamma$ is to be determined. We first note that $\psi$ is Lipschitz with constant $\max\{|\alpha - \gamma|, |\beta - \gamma|\}$. Next, we can decompose $h(x) = Hx + G\phi(Wx)$ as

$$h(x) = (H + \gamma GW)x + G\psi(Wx)$$

Now a bound on the Lipschitz constant of $h$ can be found by

$$\|h(x) - h(x')\| = \|(H + \gamma GW)(x-x') + G(\psi(Wx) - \psi(Wx'))\|$$

$$\leq \|(H + \gamma GW)(x-x')\| + \|G(\psi(Wx) - \psi(Wx'))\|$$

$$\leq \underbrace{(\|H + \gamma GW\| + \max\{|\alpha - \gamma|, |\beta - \gamma|\}\|G\|\|W\|)}_{L_1(\gamma)}\|x - x'\|.$$

While the optimal value of $\gamma$ that minimizes $L_1(\gamma)$ is not analytic in general, in special cases, we can obtain analytic solutions. We discuss one such case below.

Suppose $H = 0$. Then we can write

$$L_1(\gamma) = |\gamma|\|GW\| + \max\{|\alpha - \gamma|, |\beta - \gamma|\}\|G\|\|W\|$$

If $\gamma \leq 0$, we have

$$L_1(\gamma) = -\gamma\|GW\| + (\beta - \gamma)\|G\|\|W\| = \beta\|G\|\|W\| - \gamma(\|GW\| + \|G\|\|W\|)$$

Hence, the minimum value of $L_1(\gamma) = \beta\|G\|\|W\|$ is achieved when $\gamma = 0$. This optimal value corresponds to the naive bound.

Now if $0 \le \gamma \le (\alpha + \beta)/2$, we can write

$$L_1(\gamma) = \gamma\|GW\| + (\beta - \gamma)\|G\|\|W\| = \gamma(\|GW\| - \|G\|\|W\|) + \beta\|G\|\|W\|.$$

Since $\|GW\| \le \|G\|\|W\|$, the minimum value of $L_1(\gamma)$ is obtained when $\gamma = (\alpha + \beta)/2$.

Finally, when $\gamma \ge (\alpha + \beta)/2$, we have

$$L_1(\gamma) = \gamma\|GW\| + (\gamma - \alpha)\|G\|\|W\| = \gamma(\|GW\| + \|G\|\|W\|) - \alpha\|G\|\|W\|.$$

In this case, the optimal value occurs when $\gamma = (\alpha + \beta)/2$.

In conclusion, when $H = 0$, $\gamma = (\alpha + \beta)/2$ is the optimal solution and the corresponding optimal value is

$$L_1(\gamma = (\alpha + \beta)/2) = \frac{\alpha + \beta}{2}(\|GW\| + \|G\|\|W\|).$$

**Proof of Theorem 2**

**Theorem 2** Suppose there exists a diagonal $T$ that satisfies $G^\top G \preceq T$. Then a Lipschitz constant of (8) in $\ell_2$ norm is given by

$$L_{\text{LT-SDP}}(T) = \|H + \frac{\alpha + \beta}{2}GW\|_2 + \frac{\beta - \alpha}{2}\|W^\top TW\|_2^{\frac{1}{2}}.$$

**Proof** Note that $\psi$ is slope bounded between $-\frac{\beta - \alpha}{2}$ and $\frac{\beta - \alpha}{2}$. As a result, the formulation of LipSDP for $G\psi(Wx)$ is

$$\begin{bmatrix} \frac{(\beta - \alpha)^2}{2}W^\top TW - \rho I & 0 \\ 0 & -2T + G^\top G \end{bmatrix} \preceq 0.$$

This is equivalent to the satisfying

$$\begin{cases} \frac{(\beta - \alpha)^2}{2}W^\top TW \preceq \rho I, \\ G^\top G \preceq 2T. \end{cases}$$

By choosing a diagonal positive semidefinite $T$ that satisfies $G^\top G \preceq 2T$, the smallest feasible $\rho$ is $\rho = \frac{(\beta - \alpha)^2}{2}\|W^\top TW\|_2$. Therefore, a Lipschitz constant of $G\psi(Wx)$ is $\frac{\beta - \alpha}{\sqrt{2}}\|W^\top TW\|_2^{0.5}$. By redefining $T \leftarrow \frac{1}{2}T$, we arrive at the final statement of the theorem.

**Proof of Lemma 1**

**Lemma 1** Consider the sequences in (14). Define $\hat{H}_k := H_k + \frac{\alpha + \beta}{2}G_kW_k$ for $k = 0, \cdots, L - 1$.[6] Then

$$y_{k+1} = W_{k+1}\hat{H}_k \cdots \hat{H}_0 x_0 + \sum_{j=0}^{k} W_{k+1}\hat{H}_k \cdots \hat{H}_{j+1}G_j\psi(y_j) \quad k = 0, \cdots, L - 1. \quad (15)$$

**Proof** First, apply the loop transformation to rewrite $x_{k+1}$ as

$$x_{k+1} = H_kx_k + G_k\phi(y_k) = \underbrace{(H_k + \frac{\alpha + \beta}{2}G_kW_k)}_{\hat{H}_k} x_k + G_k\psi(y_k)$$

Since $y_k = W_kx_k$, we can write

$$\begin{aligned} y_{k+1} = W_{k+1}x_{k+1} &= W_{k+1}\hat{H}_kx_k + W_{k+1}G_k\psi(y_k) \\ &= W_{k+1}\hat{H}_k\hat{H}_{k-1}x_{k-1} + W_{k+1}G_k\psi(y_k) + W_{k+1}\hat{H}_kG_{k-1}\psi(y_{k-1}) \\ &= \cdots \\ &= W_{k+1}\hat{H}_k \cdots \hat{H}_0 x_0 + \sum_{j=0}^{k} W_{k+1}\hat{H}_k \cdots \hat{H}_{j+1}G_j\psi(y_j) \end{aligned}$$

---

[6] Note that we let $\hat{H}_k \cdots \hat{H}_{j+1}\big|_{j=k} = I$.

**Proof of Theorem 3**

**Theorem 3** Let $m_k, k \geq 1$ be defined recursively as

$$m_{k+1} = \|W_{k+1}\hat{H}_k \cdots \hat{H}_0\| + \frac{\beta - \alpha}{2} \sum_{j=0}^{k} \|W_{k+1}\hat{H}_k \cdots \hat{H}_{j+1} G_j\| m_j \tag{16}$$

with $m_0 = \|W_0\|$. Then $m_k$ is a Lipschitz constant of $x_0 \mapsto y_k, k = 0, \cdots, L-1$. In particular, $m_L$ is a Lipschitz constant of the neural network $x_0 \mapsto y_L = x_L$.

**Proof** Starting from (15), we can use the fact that a Lipschitz constant of the composition (respectively addition) of two functions is the product (respectively sum) of their Lipschitz constants:

$$\|y_{k+1} - \tilde{y}_{k+1}\|$$

$$\leq \|(W_{k+1}\hat{H}_k \cdots \hat{H}_0)(x_0 - \tilde{x}_0)\| + \sum_{j=0}^{k} \|(W_{k+1}\hat{H}_k \cdots \hat{H}_{j+1} G_j)(\psi(y_j) - \psi(\tilde{y}_j))\|$$

$$\leq \|W_{k+1}\hat{H}_k \cdots \hat{H}_0\| \|x_0 - \tilde{x}_0\| + \frac{\beta - \alpha}{2} \sum_{j=0}^{k} \|W_{k+1}\hat{H}_k \cdots \hat{H}_{j+1} G_j\| \|y_j - \tilde{y}_j\|$$

$$\leq \underbrace{\left( \|W_{k+1}\hat{H}_k \cdots \hat{H}_0\| + \frac{\beta - \alpha}{2} \sum_{j=0}^{k} \|W_{k+1}\hat{H}_k \cdots \hat{H}_{j+1} G_j\| m_j \right)}_{m_{k+1}} \|x_0 - \tilde{x}_0\|.$$

We now show that the sequence (16) produces upper bounds that are no worse than the naive upper bound.

**Proposition 4** Consider the sequence (16). Then we have

$$m_k \leq \|W_k\| \underbrace{\prod_{j=0}^{k-1} (\|H_j\| + \beta \|G_j\| \|W_j\|)}_{:=L_k}$$

where $L_k$ is the naive bound on the Lipschitz constant of $x_0 \mapsto x_k$, i.e., the product of the norms of the layers.

**Proof** To prove this proposition, we use induction on $k$. The base step of the induction is established in proposition 3. Assuming true for $k \leq K$, i.e., $m_k \leq \|W_k\| L_{k-1} \ \forall k \leq K$, we will prove $m_{K+1} \leq \|W_{K+1}\| L_K$.

First, we define $\tilde{m}_k^i$ ($1 \leq i \leq k$) as the upper bound on the Lipschitz constant of $y_k$ when we apply loop transformation to the last $i$ activation layers and then calculate an upper bound on the Lipschitz constant using the $m_j$ and $L_j, j \geq k - i$. For example, $\tilde{m}_k^1$ is when only the final activation has had the loop transformation applied on it, i.e.,

$$y_{k+1} = W_{k+1}\hat{H}_k x_k + W_{k+1} G_k \psi(y_k),$$

which yields

$$\tilde{m}_{k+1}^1 = \|W_{k+1}\hat{H}_k\| L_k + \frac{\beta - \alpha}{2} \|W_{k+1} G_k\| m_k.$$

Using the definition, it is evident that $\tilde{m}_k^k = m_k$.

$$\begin{aligned} \tilde{m}_{k+1}^0 &= \|W_{k+1} H_k\| L_k + \beta \|W_{k+1} G_k\| m_k \\ &\leq \|W_{k+1}\| \big( \|H_k\| L_k + \beta \|G_k\| m_k \big) \\ &\leq \|W_{k+1}\| \big( \|H_k\| + \beta \|G_k\| \|W_k\| \big) L_k \\ &= \|W_{k+1}\| L_{k+1} \end{aligned}$$

Next, we can now prove that $\tilde{m}_k^{i+1} \leq \tilde{m}_k^i$. For $i = 0$, the proof simply follows from proposition 3. For $i \geq 1$, consider when we have applied the loop transformation to $i$ layers. We will have an equation in the following form

$$y_k = A'_{k-i} x_{k-i} + \sum_{j=k-i}^{k-1} A_{k-j} \psi(y_{k-j}) \tag{21}$$

where the matrices $A_i$ and $A'_i$ are just used to simplify the equation instead of the actual matrices that show up after applying the loop transform in each step. We have

$$\tilde{m}_k^i = \|A'_{k-i}\| L_{k-i} + \frac{\beta - \alpha}{2} \sum_{j=k-i}^{k-1} \|A_{k-j}\| m_{k-j}.$$

Now, if we were to replace for $x_{k-i}$ and apply the loop transform one more step, instead of (21) we would have

$$y_k = A'_{k-i}\big(\hat{H}_{k-i-1} x_{k-i-1} + G_{k-i-1} \psi(y_{k-i-1})\big) + \sum_{j=k-i}^{k-1} A_{k-j} \psi(y_{k-j})$$

$$= A'_{k-i-1} x_{k-i-1} + \sum_{j=k-i-1}^{k-1} A_{k-j} \psi(y_{k-j}),$$

where we have defined $A'_{k-i-1} = A'_{k-i}\hat{H}_{k-i-1}$ and $A_{k-i-1} = A'_{k-i}G_{k-i-1}$. We then have:

$$\tilde{m}_k^{i+1} = \|A'_{k-i-1}\| L_{k-i-1} + \frac{\beta - \alpha}{2} \sum_{j=k-i-1}^{k-1} \|A_{k-j}\| m_{k-j}$$

$$\leq \|A'_{k-i}\|\Big(\|\hat{H}_{k-i-1}\| L_{k-i-1} + \frac{\beta - \alpha}{2}\|G_{k-i-1}\| m_{k-i-1}\Big)$$

$$+ \frac{\beta - \alpha}{2} \sum_{j=k-i}^{k-1} \|A_{k-j}\| m_{k-j}$$

$$\xrightarrow[\text{hypothesis}]{\text{induction}} \leq \|A'_{k-i}\|\Big(\|\hat{H}_{k-i-1}\| + \frac{\beta - \alpha}{2}\|G_{k-i-1}\|\|W_{k-i-1}\|\Big) L_{k-i-1}$$

$$+ \frac{\beta - \alpha}{2} \sum_{j=k-i}^{k-1} \|A_{k-j}\| m_{k-j}$$

$$\xrightarrow[3]{\text{proposition}} \leq \|A'_{k-i}\|\big(\|H_{k-i-1}\| + \beta\|G_{k-i-1}\|\|W_{k-i-1}\|\big) L_{k-i-1}$$

$$+ \frac{\beta - \alpha}{2} \sum_{j=k-i}^{k-1} \|A_{k-j}\| m_{k-j}$$

$$= \|A'_{k-i}\| L_{k-i} + \frac{\beta - \alpha}{2} \sum_{j=k-i}^{k-1} \|A_{k-j}\| m_{k-j} = \tilde{m}_k^i$$

As a result, $m_{K+1} = \tilde{m}_{K+1}^{K+1} \leq \tilde{m}_{K+1}^0 \leq \|W_{K+1}\| L_{K+1}$. This concludes the proof.

**Additional discussion**  For Theorem 3, we unrolled the sequence $\{y_k\}$ and then applied loop transformation on the nonlinearities. In principle, we can unroll the sequence $\{x_k\}$ and bound the Lipschitz constant of the maps $x_0 \mapsto x_k$.

**Lemma 2** The following identity holds for $x_k, k = 0, \cdots, L - 1$,

$$x_{k+1} = \hat{H}_k \cdots \hat{H}_0 x_0 + \sum_{j=0}^{k} \hat{H}_k \cdots \hat{H}_{j+1} G_j \psi(W_j x_j) \tag{22}$$

where $\hat{H}_k := H_k + \frac{\alpha+\beta}{2} G_k W_k$ for $k = 0, \cdots, L - 1$. [7]

---

[7] Note that we let $\prod_{j=k}^{k} \hat{H}_k \cdots \hat{H}_{j+1} = I$.

The proof of lemma 2 is similar to lemma 1. We are now ready to state our result for the multi-layer case.

**Theorem 4** Let $m'_k, k \geq 0$, be defined recursively as

$$m'_{k+1} = \|\hat{H}_k \cdots \hat{H}_0\|_2 + \frac{\beta - \alpha}{2} \sum_{j=0}^{k} \|W_j^\top T_{k,j} W_j\|_2^{0.5} m'_j,$$

with $m'_0 = 1$, where $T_{k,j}$ are diagonal matrices satisfying $(\hat{H}_k \cdots \hat{H}_{j+1} G_j)^\top (\hat{H}_k \cdots \hat{H}_{j+1} G_j) \preceq T_{k,j}$. Then $m'_k$ is a Lipschitz constant of $x_0 \mapsto x_k, k = 0, \cdots, L$. In particular, $m'_L$ is a Lipschitz constant of the neural network $x_0 \mapsto x_L$.

**Proof** The proof follows by induction. The base step is established in Theorem 2. Assuming true for $k \leq K$, consider eq. (22) for $k = K$ and apply LipSDP on each nonlinearity term via the introduction of diagonal matrices $T_{k,j}$. As established in theorem 2, by imposing $(\hat{H}_k \cdots \hat{H}_{j+1} G_j)^\top (\hat{H}_k \cdots \hat{H}_{j+1} G_j) \preceq T_{k,j}$, $\frac{\beta-\alpha}{2} \|W_j^\top T_{k,j} W_j\|_2^{0.5} m_j$ is a Lipschitz constant of $\hat{H}_k \cdots \hat{H}_{j+1} G_j \psi(W_j x_j)$. This concludes the proof.

**Remark** Using the recursion provided by Lemma 1, we provided Lispchitz constants $m_k$ for the map $x_0 \mapsto y_k$ in Section 3.1. We note that had we used the recursion given by Lemma 2, the resulting Lipschitz constants $\hat{m}_i$ for the map $x_0 \mapsto x_i$ would have been inferior. To see this, note that we have

$$\hat{m}_{k+1} = \|\hat{H}_k \cdots \hat{H}_0\| + \frac{\beta - \alpha}{2} \sum_{j=0}^{k} \|\hat{H}_k \cdots \hat{H}_{j+1} G_j\| \|W_j\| \hat{m}_j.$$

Note that $m_1 = \|W_1 \hat{H}_0\| + \frac{\beta-\alpha}{2} \|W_1 G_0\| m_0 = \|W_1 \hat{H}_0\| + \frac{\beta-\alpha}{2} \|W_1 G_0\| \|W_0\| \leq \|W_1\| \hat{m}_1$.

$$\hat{m}_{k+1} = \|\hat{H}_k \cdots \hat{H}_0\| + \frac{\beta - \alpha}{2} \sum_{j=0}^{k} \|\hat{H}_k \cdots \hat{H}_{j+1} G_j\| \|W_j\| \hat{m}_j.$$

Now, note that $m_0 = \hat{m}_1$. Also, since $m_1 = \|W_1 \hat{H}_0\| + \frac{\beta-\alpha}{2} \|W_1 G_0\| m_0$, we have $m_1 \leq \|W_1\| m_0 = \|W_1\| \hat{m}_1$. Using induction, we have

$$m_{k+1} = \|W_{k+1} \hat{H}_k \cdots \hat{H}_0\| + \frac{\beta - \alpha}{2} \sum_{j=0}^{k} \|W_{k+1} \hat{H}_k \cdots \hat{H}_{j+1} G_j\| m_j$$

$$\leq \|W_{k+1}\| \|\hat{H}_k \cdots \hat{H}_0\| + \frac{\beta - \alpha}{2} \sum_{j=0}^{k} \|W_{k+1}\| \|\hat{H}_k \cdots \hat{H}_{j+1} G_j\| m_j$$

$$\leq \|W_{k+1}\| \|\hat{H}_k \cdots \hat{H}_0\| + \frac{\beta - \alpha}{2} \sum_{j=0}^{k} \|W_{k+1}\| \|\hat{H}_k \cdots \hat{H}_{j+1} G_j\| \|W_j\| \hat{m}_j$$

$$\leq \|W_{k+1}\| \hat{m}_{k+1}.$$

This shows that using the modified notation provides a better Lipschitz estimate.

**Batch Normalization, Softmax, and Other Layers** The current formulation of our improved Lipschitz estimation algorithm is applicable to architectures consisting of convolutional, fully connected, and slope-restricted activation layers. Furthermore, batch normalization layers are also supported as they are affine,

$$x_i \leftarrow \gamma_i \frac{x_i - \mu_i}{\sqrt{\sigma_i^2 + \epsilon}} + \beta_i,$$

where $\sigma_i$ and $\mu_i$ are the standard deviation and the mean of the current mini-batch (or a running average), respectively, and $\gamma_i$ and $\beta_i$ are learnable parameters. These layers can essentially be treated like any other fully-connected or convolutional layer in the algorithm. Furthermore, if the batch normalization layer is next to a linear layer, then for Lipschitz calculation, it can be absorbed into that layer (as it is common to do for a fully-trained network during deployment).

As shown in [42], the softmax function is slope-restricted in $[0, 1]$. This follows from the fact that the softmax function is the gradient of the LSE function, which is convex with 1-Lipschitz gradient. As a result, softmax layers can be handled without any modification to the algorithm.

In future work, we will explore how other common layers can be embedded in our framework.

**Non-Residual Networks**    We study the improved Lipschitz estimation algorithm for the system defined in (14) in the special case in which the matrices $H_k$ and $G_k$ are zero and identity matrices of appropriate sizes, respectively. The mCnF architectures used in the experiments of this work are examples of such networks. The structure is simply given by

$$
\begin{aligned}
y_0 &= W_0 x \\
y_k &= W_k \phi(y_{k-1}), \quad k = 1, \cdots, L.
\end{aligned}
\tag{23}
$$

We have the following corollaries for this special case.

**Proposition 5** The following identity holds for $y_k, k = 1, \cdots, L-1$

$$
\begin{aligned}
y_k &= W_k \psi(y_{k-1}) + \frac{\alpha + \beta}{2} W_k W_{k-1} \psi(y_{k-2}) + (\frac{\alpha + \beta}{2})^2 W_k W_{k-1} W_{k-2} \psi(y_{k-3}) \\
&+ \cdots + (\frac{\alpha + \beta}{2})^{k-1} W_k \cdots W_1 \psi(y_0) + (\frac{\alpha + \beta}{2})^k W_k \cdots W_0 x
\end{aligned}
$$

As a result, we have the following proposition.

**Proposition 6** Let $m_k$ be a Lipschitz constant of $x \mapsto y_k, k = 1, \cdots, L$. Then

$$
m_k = \frac{\beta - \alpha}{2} \sum_{i=1}^{k} (\frac{\alpha + \beta}{2})^{k-i} \|W_k \cdots W_i\| m_{i-1} + (\frac{\alpha + \beta}{2})^k \|W_k \cdots W_0\|.
\tag{24}
$$

with $m_0 = \|W_0\|$. In particular, $m_L$ is a Lipschitz constant of the neural network $x \mapsto y_L$.

The proofs for these special cases are similar to their general cases.

**Comparison with [56]**    For the special case of non-residual networks, i.e., $H_i = 0, \forall i$, and when $\alpha = 0$ and $\beta = 1$, our result recovers Proposition 4.3.iv of [56]. To see this, define

$$
J_{k,i} = \{(j_1, \cdots, j_i) | 0 < j_i < \cdots < j_1 \leq k\}
$$

$$
(j_1, \cdots, j_i) \in J_{k,i} : \sigma_{k,(j_1, \cdots, j_i)} = \|W_k \cdots W_{j_1}\| \|W_{j_1 - 1} \cdots W_{j_2}\| \cdots \|W_{j_i - 1} \cdots W_0\|
$$

with $\sigma_{k,\varnothing} = \|W_k \cdots W_0\|$.

**Proposition 7** Let $\alpha = 0$ and $\beta = 1$ and consider $m_k$ as in Proposition 6. Define $\theta_k$ as follows

$$
\theta_k = \frac{1}{2^k} \Big( \|W_k \cdots W_0\| + \sum_{i=1}^{k} \sum_{(j_1, \cdots, j_i) \in J_{k,i}} \sigma_{k,(j_1, \cdots, j_i)} \Big).
$$

Then $\theta_k = m_k$.

**Proof** Proof by induction. The base step is easily verified by checking that

$$
\theta_1 = \frac{1}{2} (\|W_1 W_0\| + \|W_1\| \|W_0\|) = \frac{1}{2} (\|W_1 W_0\| + \|W_1\| m_0) = m_1.
$$

Assume true by the induction hypothesis that $\theta_k = m_k$ for $k \leq K - 1$. We have

$$
\theta_K = \frac{1}{2^K} \Big( \|W_K \cdots W_0\| + \sum_{i=1}^{K} \sum_{(j_1, \cdots, j_i) \in J_{K,i}} \sigma_{K,(j_1, \cdots, j_i)} \Big)
$$

$$
= \frac{1}{2^K} \Big( \|W_K \cdots W_0\| + \sum_{i=1}^{K} \sum_{j_1=i}^{K} \sum_{(j_2, \cdots, j_i) \in J_{j_1-1,i-1}} \sigma_{K,(j_1, \cdots, j_i)} \Big)
$$

$$
= \frac{1}{2^K} \Big( \|W_K \cdots W_0\| + \sum_{i=1}^{K} \sum_{j_1=i}^{K} \sum_{(j_2, \cdots, j_i) \in J_{j_1-1,i-1}} \|W_K \cdots W_{j_1}\| \sigma_{j_1-1,(j_2, \cdots, j_i)} \Big)
$$

$$
= \frac{1}{2^K} \Big( \|W_K \cdots W_0\|
$$

$$
+ \sum_{j_1=1}^{K} \|W_K \cdots W_{j_1}\| \underbrace{ \Big( \sigma_{j_1-1,\varnothing} + \sum_{i=2}^{j_1} \sum_{(j_2, \cdots, j_i) \in J_{j_1-1,i-1}} \sigma_{j_1-1,(j_2, \cdots, j_i)} \Big) }_{\theta_{j_1-1}} \Big) \text{(Interchange sums)}
$$

$$
= \frac{1}{2^K} \Big( \|W_K \cdots W_0\| + \sum_{j_1=1}^{K} \|W_K \cdots W_{j_1}\| m_{j_1-1} \Big) = m_K.
$$

It is straightforward to extend our formulation to also derive Theorem 4.2 of [56].

Compared to [56], our derivation lends itself to efficient implementation due to its recursive nature. Furthermore, as our algorithm calculates the Lipschitz constants of all principal subnetworks, i.e., the maps $x_0 \mapsto y_k, k = 0, \cdots, L$, it saves on a lot of repeated computations for applications that require Lipschitz constants of all principal subnetworks. Finally, our method can be easily adapted to compute local Lipschitz constants. We will explore these aspects in future works.

## A.4  Implementation

Theorem 3 states a method to calculate an upper bound on the Lipschitz constant of a general neural network in any norm. To implement the algorithm, we would need to calculate a multitude of matrix-matrix multiplications and then compute their norms. This can be computationally expensive when large matrices are involved, especially convolution layers which require the formation of the equivalent Toeplitz matrix. This seems to limit the application of the method to small networks without convolutional layers, but using the power iteration, we can circumvent the need for calculating matrix-matrix multiplications by performing much more inexpensive matrix-vector multiplications. We acknowledge that this efficient implementation is only applicable to $\ell_2$ norms. For future work, we will investigate if similar tricks can be used for other norms such as $\ell_\infty$.

**Power Iteration**  Calculating the $\ell_2$ norm of a general rectangular matrix can be performed by using the *power iteration* [25]. For a given linear layer with weight $W$, or convolutional layer with equivalent Toeplitz weight matrix $W$ (refer to [25] for details on how to construct $W$ from the kernels), the power method update rule is given by $x \leftarrow W^\top W x$. As stated in [25], for networks consisting of linear and convolutional layers, the operation $W^\top W x$ is equivalent to first a forward propagation of $x$ through the layer with weight $W$, and then a backward propagation through the same layer. As a result, calculating the Lipschitz constant amounts to a set of forward and backward propagations through the layers.

To multiply several matrices, it suffices to consider the sequential linear network constructed by layers corresponding to the matrices and performing forward and backward passes on this network, as outlined in Algorithm 1[8]. Furthermore, building on the observations of [5, 20], if we save the vector

---

[8]The indexing $j = m, \cdots, 1$ means that the variable $j$ starts from $m$ and decreases one by one until it reaches 1.

$x$ in Algorithm 1 in memory during training, then the number of power iterations (the parameter $N$) does not need to be large. This argument is based on the fact that after each iteration of training, the change in the weights of the network is relatively small, and performing only a few iterations of the power method would suffice throughout the overall training process.

**Efficient Computation of Pair-wise Lipschitz Bounds**   The derived lower bound (4) on the certified radius requires the calculation of the $K - 1$ Lipschitz constants $L_{yi}, \forall i \neq y$ for each data point $x$ having label $y$ (recall that $L_{yi}$ is the global Lipschitz constant of $z_y(x; \theta) - z_i(x; \theta)$). As $L_{yi}$ is a global constant, it can be used for any data point $x'$ with the same label $y$. Since each mini-batch of the training data typically contains data points from all classes (especially on smaller datasets like MNIST and CIFAR-10), for each mini-batch, we calculate all possible pairwise Lipschitz constants $L_{ij}, i \neq j$ (a total of $K(K - 1)/2$ constants[9]), and then, compute the lower bounds for all the data points in the mini-batch.

We note that $L_{ij} \leq \sqrt{2}L$ and $L_{ij} \leq L_i + L_j$, where $L$ and $L_i$ are the Lipschitz constants of the network (the logit vector $z(x; \theta)$) and the $i$-th logit ($z_i(x; \theta)$), respectively. These upper bounds would reduce the number of Lipschitz calculations in each mini-batch from $K(K - 1)/2$ to 1 and $K$, respectively, as pursued in prior work [5, 19]. However, these upper bounds would be relatively loose, as reported in Table 2. In the following, we will discuss our approach to computing the $L_{ij}$s more efficiently.

Consider the network structure (14). As $L_{ij}$ is the Lipschitz constant of $z_i(x; \theta) - z_j(x; \theta) = (u_i - u_j)^\top z(x; \theta)$, and as $z(x; \theta) = y_L = x_L = H_{L-1}x_{L-1} + G_{L-1}\phi(y_{L-1})$, we have

$$(u_i - u_j)^\top z(x; \theta) = \underbrace{((u_i - u_j)^\top H_{L-1})}_{H_{ij}} x_{L-1} + \underbrace{((u_i - u_j)^\top G_{L-1})}_{G_{ij}} \phi(y_{L-1}) \qquad (25)$$

As a result, calculating $L_{ij}$ amounts to calculating the Lipschitz constant of a network whose last layer's weights have been modified to $H_{ij}$ and $G_{ij}$. As Eq. (25) shows, the difference between $L_{ij}$ and $L_{kl}$ is simply in the last layer. Using the naive Lipschitz bound, i.e.,

$$L_{ij} = (\|H_{ij}\| + \beta\|G_{ij}\|\|W_{L-1}\|) \prod_{k=0}^{L-2} (\|H_k\| + \beta\|G_k\|\|W_k\|),$$

it is evident that the calculation for different pairs has much in common and we can cache the common term in memory and prevent redundant computations.

---

[9]Note that $L_{ij} = L_{ji}$

---

**Algorithm 1** Multi-matrix power iteration.

---

**Input** Matrices $A_1, \cdots, A_m$, number of power iterations $N$.
**Output** The spectral norm $\|A_m A_{m-1} \cdots A_1\|_2$.
Randomly initialize $x$.
**for** $i = 1, \cdots, N$ **do**
    **for** $j = 1, \cdots, m$ **do**                             ▷ Perform forward pass
        $x \leftarrow A_j x$
    **end for**
    **for** $j = m, \cdots, 1$ **do**                           ▷ Perform backward pass
        $x \leftarrow A_j^\top x$
    **end for**
    $x \leftarrow \dfrac{x}{\|x\|_2}$
**end for**
**for** $j = 1, \cdots, m$ **do**
    $x \leftarrow A_j x$
**end for**
**return** $\|x\|_2$

---

A careful analysis of Eq. (16) reveals that the previous structure is also present in our improved method and the values of $m_k$ are the same for all $k = 0, \cdots, L-1$ and only $m_L$ changes for different pairs of $(i, j)$. Given that we defined $W_L = I$, the formulation for the pairwise Lipschitz constants using our improved method is

$$L_{ij} = \|(u_i - u_j)^\top \hat{H}_{L-1} \cdots \hat{H}_0\| + \frac{\beta - \alpha}{2} \sum_{j=0}^{L-1} \|(u_i - u_j)^\top \hat{H}_{L-1} \cdots \hat{H}_{j+1} G_j\| m_j \qquad (26)$$

There is still one more aspect of the Lipschitz calculation algorithm (16) that we can exploit, which due to parallel GPU-based implementation, reduces the practical run time to that of the naive estimation algorithm. We will discuss this next.

Since our experiments are only of the simpler structure (23), and as the explanation of the parallel implementation is simpler on this structure, we provide the details for this case here. Extrapolating these details to the more general structure (14) follows a similar framework and the same general idea. To better portray the architecture that we can utilize for parallel implementation, let's consider a neural network with $L = 3$ and consider the computations required to calculate $m_0, m_1$, and $m_2$ (the calculation of $m_3$ will use another observation that does not completely need this portion of the parallel implementation). We have the following

$$m_0 = \|W_0\|$$

$$m_1 = \frac{\beta - \alpha}{2} \|W_1\| m_0 + \|W_1 W_0\|$$

$$m_2 = \frac{\beta - \alpha}{2} \|W_2\| m_1 + \frac{\beta - \alpha}{2} \|W_2 W_1\| m_0 + \|W_2 W_1 W_0\|$$

As we will use the power method, let $x_{ij}, i = 0, \cdots, L - 1, j = 0, \cdots, i$ be the (Eigen)vector saved for the power method of calculating the $j$th norm from the end in the $i$th equation, i.e., $x_{00}, x_{10}, x_{11}, x_{20}, x_{21}, x_{22}$ are the (eigen)vectors used for $\|W_0\|, \|W_1 W_0\|, \|W_1\|, \|W_2 W_1 W_0\|, \|W_2 W_1\|$, and $\|W_2\|$, respectively. We can then note that concerning Algorithm 1, we will need to perform a forward pass of $x_{00}, x_{10}$, and $x_{20}$ through $W_0$. Then we need to perform a forward pass of $(W_0 x_{10}), (W_0 x_{20})^{10}, x_{11}$, and $x_{21}$ through $W_1$. Finally, we need to perform a forward pass of $(W_1(W_0 x_{20})), (W_1 x_{21})$, and $x_{22}$ through $W_2$. Then, similarly, we need to perform backward passes where we can use the same kind of batched inputs, i.e., perform a backward pass of $(W_2(W_1(W_0 x_{20}))), (W_2(W_1 x_{21}))$, and $(W_2 x_{22})$ through $W_2$, a backward pass of $(W_1(W_0 x_{10})), (W_2^\top(W_2(W_1(W_0 x_{20})))), W_1 x_{11}$, and $(W_2^\top(W_2(W_1 x_{21})))$ though $W_1$, and so on. Performing this for $N$ times satisfies the first for-loop of Algorithm 1. Figure 3 portrays the batch implementation for this toy example. It then suffices to do another forward pass of this scheme and extract the norms. We present this procedure in Algorithm 2 [11] [12] [13] [14] [15].

The final important observation is that calculating the pairwise Lipschitz constant $L_{ij}$ amounts to calculating the Lipschitz constant of a network with final layer weight $(u_i - u_j)^\top W_L$, which is a $1 \times n_{L-1}$ matrix. Therefore, $(u_i - u_j)^\top W_L \cdots W_k$ will also be a matrix of similar dimensions $1 \times n$ for a suitable $n$. The calculation of the $\ell_2$ norm of such a matrix amounts to calculating the $\ell_2$

---

[10]The parentheses are shown to emphasize the order of operations.

[11]We use the `Python` language standard for indexing. Indexing starts from 0, and negative indices iterate through the object from the end. Note that all indexing in the algorithm only considers the batched dimension and does not affect the data axes.

[12]The function `Concat` takes as input two $x$ and $y$ and concatenates them in the batch dimension such that the index corresponding to $y$ comes after the index for $x$.

[13]In this algorithm we have used the notation of $Wx$ and $W^\top x$ for applying a layer with weight on $x$ on a forward and backward pass, respectively, for simplicity. As proven in [25], for convolutional layers, these operations are equivalent to simply applying `Conv` and `Conv_Transpose` with the weight $W$. The functions for these transforms are readily available in deep learning libraries such as `PyTorch`

[14]Although Algorithm 2 uses many nested for-loops, in practice, many of the for-loops are handled by being able to index many objects at once. For example, in performing the forward pass, the two nested for-loops for variables $j$ and $k$ are handled using a single call on the variable $y_i$.

[15]As it can be inferred from [25], the calculation of the $\ell_2$ norm of a non-batched tensor $x$ with more than one dimension is performed by flattening the vector to a single dimensional vector and then taking the $\ell_2$ norm.

**Algorithm 2** Improved Lipschitz Batched Implementation

---

**Input** Layer weights $W_0, \cdots, W_{L-1}$, number of power iterations $N$, saved (eigen)vectors $x_{ij}, i = 0, \cdots, L-1, j = 0, \cdots, i$.
**Output** $\|W_i \cdots W_j\|$ for $i = L-1, \cdots, 0, j = i, \cdots, 0$
**for** pCounter $= 1, \cdots, N$ **do**
    $x = []$                                            ▷ Initialize power iteration variable
    **for** $i = 0, \cdots, L-1$ **do**                          ▷ Perform forward passes
        **for** $j = 0, \cdots, i-1$ **do**
            **for** $k = 1, \cdots, L-1-i$ **do**
                $x \leftarrow \mathrm{Concat}([x, y_i[j * (L - i + 1) + k]])$    ▷ Add vectors that need to continue
            **end for**
        **end for**
        **for** $j = i, \cdots, L-1$ **do**
            $x \leftarrow \mathrm{Concat}([x, x_{ji}]$
        **end for**
        $y_{i+1} = W_i x$
    **end for**                                    ▷ Finished forward passes

    **for** $i = L-1, \cdots, 0$ **do**                       ▷ Perform backward passes
        $y_i' = W_i^\top y_i$
        **for** $j = L-1, \cdots, i$ **do**
            $x_{j,i} = y_i'[-1 - (L - 1 - j)]$         ▷ Remove finished components
        **end for**
        **for** $j = 0, \cdots, i-1$ **do**
            **for** $k = 1, \cdots, L-1-i$ **do**
                 $y_{i-1}[j * (L - i + 1) + k] = y_i'[j * (L - i) + k - 1]$ ▷ Update $y_{i-1}$ for next step of
backward pass
            **end for**
        **end for**
    **end for**                                    ▷ Finished backward passes
    Normalize every $x_{ij}$ in $\ell_2$ norm.
**end for**                           ▷ Finish update of (eigen)vectors using power iteration

$x = []$                                      ▷ Final forward pass to extract norms
**for** $i = 0, \cdots, L-1$ **do**
    **for** $j = 0, \cdots, i-1$ **do**
        **for** $k = 1, \cdots, L-1-i$ **do**
            $x \leftarrow \mathrm{Concat}([x, y_i[j * (L - i + 1) + k]])$    ▷ Add vectors that need to continue
        **end for**
    **end for**
    **for** $j = i, \cdots, L-1$ **do**
        $x \leftarrow \mathrm{Concat}([x, x_{ji}]$
    **end for**
    $y_{i+1} = W_i x$
    **for** $j = 0, \cdots, i$ **do**
        $\|W_i \cdots W_j\|_2 \leftarrow \|y_{i+1}[j * (L - i)]\|_2$         ▷ Extract norms
    **end for**
**end for**

---

norm of the vector $W_k^\top \cdots W_L^\top (u_i - u_j)$. Comparing this with the backward passes mentioned in Algorithm 1, it is clear that this amounts to passing the vector $u_i - u_j$ backward from the weights $W_L, \cdots, W_k$. This also abolishes the need for performing the power iteration for computations involving the last layer. We present the final algorithm in Algorithm 3 [16]. Our implementation utilizes

---

[16]Note that Algorithm 3 can also be implemented in a batched format as the two outer for-loops can be removed by introducing a proper $x$. As this is similar to the idea in Algorithm 2, we have left it out to simplify the exposition.

---

**Algorithm 3** Pairwise improved Lipschitz calculation

---

**Input** Layer weights $W_0, \cdots, W_L$, number of classes $K$, number of power iterations $N$, saved (eigen)vectors $x_{ij}, i = 0, \cdots, L-1, j = 0, \cdots, i$.
**Output** Pairwise Lipschitz constants $L_{ij}$
Calculate the values $m_i, i = 0, \cdots, L-1$ by using Algorithm 2 on the weights $W_0, \cdots, W_{L-1}$ with (eigen)vectors $x_{ij}, i = 0, \cdots, L-1, j = 0, \cdots, i$ and then using equation (24).

Initialize $x = []$
**for** $i = 1, \cdots, K-1$ **do**
    **for** $j = i+1, \cdots, K$ **do**
        $x \leftarrow \text{Concat}([x, u_i - u_j]$
    **end for**
**end for**
**for** $k = L, \cdots, 0$ **do**
    $x \leftarrow W_k^\top x$
    batchIndex $\leftarrow 0$
    **for** $i = 1, \cdots, K-1$ **do**
        **for** $j = i+1, \cdots, K$ **do**
            $\|(u_i - u_j)^\top W_L \cdots W_k\|_2 \leftarrow \|x[\text{batchIndex}]\|_2$
            batchIndex $\leftarrow$ batchIndex + 1
        **end for**
    **end for**
**end for**
Calculate pairwise Lipschitz constants using (26)

---

all the mentioned points.

**Computational Complexity** To ease exposition, we state the arguments for the simpler case of calculating the Lipschitz constant of the whole network of architecture (23) and then extend them to pairwise Lipschitz calculation. The analysis can be extrapolated to the residual architecture of the paper.

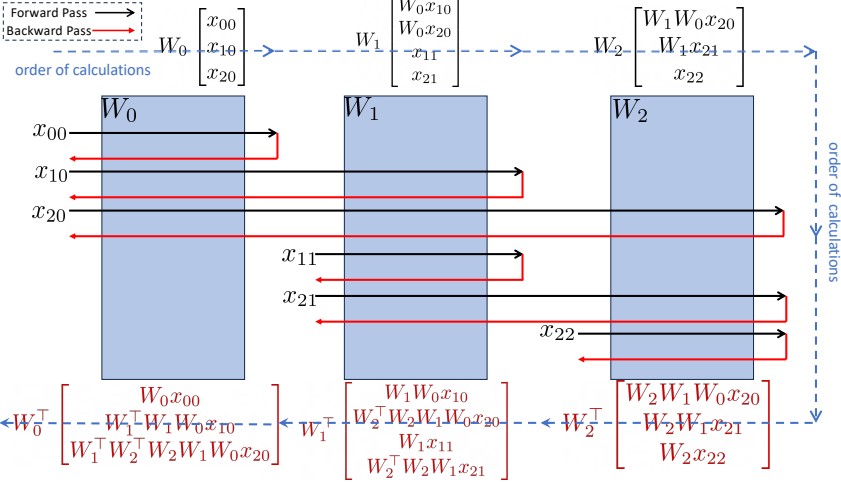

Figure 3: Illustration of using mini-batching for parallelizing the power iteration process of our proposed algorithm on the proposed toy example. The black lines and equations represent the calculations for the forward propagation and the red are for the backward propagation. In this figure, the vertical stacking operation $\begin{bmatrix} x \\ y \end{bmatrix}$ represents concatenation in the mini-batch dimension.

For the naive Lipschitz estimation method, we need to calculate $L + 1$ matrix norms $\|W_i\|, i = 0, \cdots, L$. To calculate each norm using the power method, we need to calculate $W^\top W x$ $N$ times and then calculate $Wx$. This results in $2N + 1$ forward passes through either $W$ or $W^\top$. As a result, for the full network, this would simply become $(L + 1)(2N + 1)$ passes through different weights. As our method requires the norm calculation of matrices that may be the multiplication of several matrices, to provide the same complexity we need to count the number of matrices that show up for each $m_i$. It is easy to see that if $m_i, i < k$ are calculated, then to calculate $m_k$ one needs to calculate $k + 1$ new matrix norms in which a total of $\frac{(k+1)(k+2)}{2}$ matrices appear. For example, $m_2$ includes the terms $W_2$, $W_2 W_1$, and $W_2 W_1 W_0$, which totals 6 matrices. Put together, we see that the complexity, in terms of the number of passes through different weights, is

$$\sum_{i=0}^{L}(2N + 1)\frac{(i+1)(i+2)}{2} = \frac{(2N+1)(L+1)(L+3)(L+5)}{6}. \tag{27}$$

We note that extending the analysis to pairwise Lipschitz calculation is straightforward. For the naive Lipschitz estimation algorithm, instead of $(L+1)(2N+1)$ passes, we will require $L(2N+1) + \binom{k_m}{2}$ passes through different weights, where $k_m$ is the number of classes in the current mini-batch. If we need to calculate all the pairwise Lipschitz constants then $k_m$ is equal to the total number of classes. The reason for this change is that the pairwise Lipschitz calculation only modifies the last weight of the network, i.e., to calculate $L_{ij}$ we essentially modify the last weight of the network to become $(u_i - u_j)^\top W_L$. But as all the previous calculations are the same for different pairs and since the calculation of $\|(u_i - u_j)^\top W_L\|$ only involves passing the vector $u_i - u_j$ backward through the weight $W_L$, the new calculations show up as additive terms. Similarly, for LipLT, the summation in (27) would be up until $L - 1$ instead of $L$ and then, $\binom{k_m}{2}\frac{(L+1)(L+2)}{2}$ extra backward passes are required through different weights.

**Time Complexity**   If parallelization is not possible, then the computational complexities presented in the previous paragraph also represent the time complexities. However, if we use the parallelization that GPUs enable, so that we can propagate several vectors through a weight in a single time unit, then using the recurrent structure in the algorithm, we can obtain much better time complexities. Here, we assume that calculating $Wx$ requires a single time unit.

Based on the explanation for Alg. 2, if we were to use the algorithm for the whole network, we find that the number of passes through different weights is the same as the naive Lipschitz estimation algorithm. We only need to take into account the time required for indexing and concatenation. Based on Alg. 2, for the forward pass of weight $W_i$, we first need to concatenate vectors that are continuing from the previous weight $W_{i-1}$ to the new vectors that have to also pass $W_i$, and after calculating $W_i x$, we need to index the vectors that do not need to pass through the following $W_{i+1}$ and concatenate them. Following the logic of the algorithm, it can be seen that for weight $W_i$, $(i + 1)(L + 1)$ vectors need to be concatenated, and after passing $W_i$, $(i + 1)$ will be indexed and concatenated as they do not need to pass $W_{i+1}$. The backward pass is similar. So $\mathcal{O}(\sum_{i=0}^{L}(i + 1)(L + 2)) = \mathcal{O}(L^3)$ concatenations are needed. However, as the time required for a pass through a weight matrix is far larger than the time required for indexing and concatenation, the time complexity remains $\mathcal{O}(LN)$; this is the same as the naive Lipschitz estimation algorithm.

For the pairwise Lipschitz calculations, given that all the $m_i, i < L$ have been computed, we only need to pass $\binom{k_m}{2}$ vectors backward through the last layer or all the layers for the naive method or LipLT, respectively. Thus the respective time complexities, using GPU parallelization, are $\mathcal{O}(LN + \binom{k_m}{2})$ and $\mathcal{O}(LN + L\binom{k_m}{2})$.

**Techniques to Reduce Time Complexity**   Although the parallel implementation provides a significant boost for the LipLT algorithm, having too many classes can nonetheless impose a heavy load on the implementation. For example, in our experiments for Tiny-Imagenet, we let $N = 1$, but the number of total pairwise Lipschitz constants is around $20,000$. As a result, the term $\mathcal{O}(L\binom{k_m}{2})$ dominates the complexity. In here we simply outline several simple workarounds to avoid this surge in complexity. Our proposals are as follows:

1. Network Lipshcitz constant: Calculate the Lipschitz constant $M$ of the whole network, and then note that $L_{ij} \leq \sqrt{2}M$. As discussed in the preceding, the time complexity of the parallel implementation is $\mathcal{O}(LN)$.

2. Per class Lipschitz constants: Calculate the Lipschitz constant of each class $L_i$ and then note that $L_{ij} \leq L_i + L_j$. Similar to the implementation of the pairwise Lipschitz, the per-class Lipschitz calculation also only modifies the last layer. As a result, with the parallel implementation, the time complexity becomes $\mathcal{O}(LN + LK)$.

3. Last layer naive Lipschitz constants: Calculate the $\hat{M}$ Lipschitz constant of the network until the penultimate layer using LipLT. View the last layer, with the corresponding vector multiplication, i.e., $(u_i - u_j)^\top W_L$ as a separate function, and multiply the Lipschitz constants. Then $L_{ij} \leq \hat{M}\|(u_i - u_j)^\top W_L\|$. The time complexity then becomes the same as the naive implementation $\mathcal{O}(LN + \binom{k_m}{2})$.

The above methods provide ways to further overapproximate the pairwise Lipschitz constants to save time. There is currently another level of redundancy that we have not exploited and we can further modify the implementation of the training algorithm to take advantage of it. As mentioned previously, there are $\binom{k_m}{2}$ pairwise Lipschitz constants that need to be calculated for a given mini-batch having $k_m$ different labels. For smaller datasets such as MNIST or CIFAR-10, each mini-batch will probably have all different $K$ labels. However, for even slightly larger datasets this is not the case. That is, as long as the batch size is less than the number of classes, then each mini-batch will only have a subset of the different labels. For example, for Tiny-Imagenet, if we choose the common batch size of 128, then each mini-batch will only contain at most 128 different labels. As a result, with correct implementation, we would only need to calculate $\binom{k_m}{2}$ pairwise Lipschitz constants, rather than $\binom{K}{2t}$. Following this observation, we propose a modification to the data loaders. The data loaders would choose a smaller subset of all the labels, and then provide $k_m$ random samples from those labels. It may be of concern whether this will alter the training and the convergence. We will investigate this in future works.

# B Experiments

## B.1 Experiment Details

In this section, we will introduce the models, hyperparameters, and further details of the implementation for § 4. We conducted the experiments on a single NVIDIA A100 GPU with 40GB of RAM [17]. For the ablation studies of this section, we fix a single random seed and report the results.

### B.1.1 $\ell_2$ Robustness

**CRM Hyperparameters** For all datasets we use $g(x, y; \theta, t) = \begin{cases} -\underline{R}_t^{\text{soft}}(x, y; \theta) & \underline{R}(x, y; \theta) \leq r_0 \\ 0 & \underline{R}(x, y; \theta) > r_0 \end{cases}$, where $t$ is the smoothing parameter and $r_0$ is a truncation parameter . The choice of $(t, r_0, \lambda)$ for MNIST, CIFAR-10, Tiny-Imagenet are $(5, 2.2, 30), (5, 0.2, 15)$, and $(12, 0.2, 150)$, respectively, where $\lambda$ is the regularization constant.

Taking advantage of the full capacity of CRM requires the calculation of pairwise Lipschitz constants between any pair of classes. For MNIST and CIFAR-10 we use LipLT to do so. For Tiny-Imagenet, as the number of classes is relatively large, bounding the pairwise Lipschitz would be computationally heavy. To mitigate this, we use LipLT up until the penultimate layer and then use the naive Lipschitz estimate for the last linear layer.

---

[17]Hosted by the Advanced Research Computing at Hopkins (ARCH) core facility (rockfish.jhu.edu), supported by the National Science Foundation (NSF) grant number OAC1920103.

**Architectures**  We use architectures of the form $mCnF$, where $m$ indicates the number of convolutional layers and $n$ is the number of fully connected layers that follow. Each layer is followed by an element-wise ReLU activation function. Convolutional layers are of the form `C(c, k, s, p)`, where `c` is the number of filters, `k` is the size of the square kernel, `s` is the stride length, and `p` is the symmetric padding. Fully connected layers are of the form `L(n)`, where `n` is the number of output neurons of this layer. The used architectures are as follows:

- 4C3F: `C(32, 3, 1, 1), C(32, 4, 2, 1), C(64, 3, 1, 1), C(64, 4, 2, 1), L(512), L(512), L(10)`
- 6C2F: `C(32, 3, 1, 1), C(32, 4, 2, 1), C(64, 3, 1, 1), C(64, 4, 2, 1), C(64, 3, 1, 1) C(64, 4, 2, 1), L(512), L(10)`
- 8C2F: `C(64, 3, 1, 1), C(64, 3, 1, 0), C(64, 4, 2, 0), C(128, 3, 1, 1), C(128, 3, 1, 1), C(128, 4, 2, 0), C(256, 3, 1, 1), C(256, 4, 2, 0), L(256), L(200)`

We note that the 4C3F and 6C2F architectures above are the same as those defined in [20]. However, the 8C2F architecture is slightly modified in the padding numbers (the number of filters and neurons are unchanged) to accommodate the LipLT algorithm.

**Optimizer**  We use the Adam optimizer [68] in the `PyTorch` library. We set the following parameters as constants for the optimizer: $\beta_1, \beta_2 = (0.9, 0.999)$, `amsgrad: False`, `eps` $= 10^{-7}$. For the learning rate, we start with a learning rate $\text{lr}_0$ and train with a constant learning rate until epoch `lr_decay` and then decay the learning rate by an appropriate multiplier $\gamma$ such that it reaches a final learning rate $\text{lr}_f$ on the last epoch of the training. We denote this schedule as $(\text{lr}_0, \text{lr}_f, \text{lr\_decay})$.

**Power Method**  Following the arguments in [5, 20, 23], using saved initializations of the (eigen)vectors for the power method, only a small number of N iterations are required for each call to the power method. The choice is usually set as values from $\{1, 2, 5, 10\}$. The hyperparameter *power iterations* in Table 3 refers to the choice of this value. We study the effects of this parameter in Table 4. For each dataset in this table, we fix a hyperparameter setting and only change the number of power iterations in different runs.

**Warm-up**  We empirically found that it is necessary to have an initial round of normal (non-regularized) training in which the network is simply trained to achieve higher clean accuracies. We found that if the model was not allowed to reach an initial non-random clean accuracy before the onset of robust training, the model would either train poorly to a much lower accuracy or not train

Table 3: Choice of hyperparameters for different architectures and datasets.

|  | 4C3F | 6C2F | 8C2F |
|---|---|---|---|
| Learning rate | $(10^{-3}, 10^{-8}, 200)$ | $(10^{-3}, 10^{-6}, 200)$ | $(10^{-4}, 10^{-6}, 100)$ |
| Batch size | 512 | 512 | 256 |
| Epochs | 500 | 400 | 300 |
| Power iterations | 10 | 5 | 5 |
| Warm-up | 1 | 10 | 10 |
| Data augmentation | $(0, 0)$ | $(5, 0.1)$ | $(20, 0.2)$ |

Table 4: Effect of the number of power iterations on accuracy.

| Number of power | MNIST (4C3F) | | | CIFAR-10 (6C2F) | | | Tiny-Imagenet (8C2F) | | |
|---|---|---|---|---|---|---|---|---|---|
| iterations N | Clean | PGD | Certified | Clean | PGD | Certified | Clean | PGD | Certified |
| 1 | 95.39 | 86.25 | 57.94 | 74.04 | 71.58 | 63.27 | 22.34 | 21.76 | 16.78 |
| 2 | 95.54 | 86.49 | 59.15 | 74.86 | 71.97 | 63.72 | 24.4 | 23.38 | 17.94 |
| 5 | 95.91 | 86.95 | 61.64 | 74.82 | 72.31 | 64.16 | 23.97 | 23.04 | 17.98 |
| 10 | 95.98 | 87.15 | 61.98 | 74.13 | 71.57 | 63.77 | 23.84 | 22.44 | 17.66 |

at all. On the other hand, we also noticed that in certain scenarios having a long initial warm-up phase could also prevent the robust training altogether, i.e., although the robust loss value would be significant in comparison to the normal loss, the training process could not increase the verified accuracy of the model. We found that a small warm-up of around 10 epochs is enough. For MNIST, as the model would reach accuracies of more than 90% in a single epoch, we only perform a single epoch of warm-up training.

**Data Augmentation**   Using a similar strategy as previous works [5, 20], we perform data augmentation on CIFAR-10 and Tiny-Imagenet datasets. We only perform rotations and translations. We say the data augmentation parameters are $(d, t)$ if every data point is randomly rotated between $[-d, d]$ degrees and translated by a ratio of $t$ in each axis. For implementation, we use the `torchvision.transforms.RandomAffine` function of the `PyTorch` library.

**PGD**   To evaluate the adversarial accuracy of the different models we use the method of [4]. We run the algorithm for 100 iterations with a constant step size of 0.001.

### B.1.2   Ablation Study

Here we present the results of different runs with several hyperparameters. The results for MNIST, CIFAR-10, and Tiny-Imagenet are provided in Tables 5, 6, and 7, respectively.
One main takeaway from these experiments is that for a given value of the smoothness parameter $t$, if the regularization parameter $\lambda$ is not large enough, the model only trains for normal accuracy and does not present any robustness guarantees and fails with a verified accuracy of 0.

An important parameter that we study is the choice of $r_0$. The effects of changing this parameter are portrayed in the aforementioned tables. To provide further insight, we plot the histogram of the certified radii in Fig. 4 for MNIST and CIFAR-10. As $r_0$ increases, the certified radius tends to increase, and this effect is best portrayed for the results on CIFAR-10, i.e., Fig. 4b. But as the results in Tables 5 and 6 show, this increase in certified radii may be accompanied by a decrease in the clean accuracy of the model due to the trade-off between the two goals of robustness and accuracy.

**Effect of Different Components**   We study the effect of the different components of the algorithm on the training of CIFAR-10. Table 8 reflects these experiments. We consider the effect of different proxies for the pairwise Lipschitz calculation, i.e., using per class Lipschitz constant or the network's Lipschitz constant to acquire an upper bound the pairwise Lipschitz constants (the rows of tab. 8), the effectiveness of the new LipLT method versus the naive method (1st vs 3rd column of tab. 8), and the effect of using the soft lower bound $\underline{R}_t^{soft}(x, y; \theta)$ versus the hard lower bound $\underline{R}(x, y; \theta)$ (1st vs 2nd column). For the hard radius experiment, we further alter the choice of the function $h$

Table 5: Sensitivity to CRM hyperparameters on MNIST (4C3F)

|  | Hyperparameter setting | Clean (%) | PGD (%) | Certified (%) |
|---|---|---|---|---|
| Original setting | $t = 5, \lambda = 30, r_0 = 2.2$ | 95.98 | 87.15 | 61.98 |
| Effect of $\lambda$ | $t = 5, \lambda = 40, r_0 = 2.2$ | 95.98 | 86.76 | 62.23 |
|  | $t = 5, \lambda = 20, r_0 = 2.2$ | 96.26 | 88.46 | 62.4 |
| Effect of $r_0$ | $t = 5, \lambda = 30, r_0 = 1.6$ | 97.33 | 90.16 | 59.33 |
|  | $t = 5, \lambda = 30, r_0 = 2.0$ | 96.42 | 88.19 | 61.64 |
|  | $t = 5, \lambda = 30, r_0 = 2.5$ | 95.4 | 85.76 | 61.41 |
| Effect of $t$ | $t = 3, \lambda = 15, r_0 = 2.2$ | 95.6 | 86.84 | 61.07 |
|  | $t = 3, \lambda = 3, r_0 = 2.2$ | 96.31 | 87.62 | 60.7 |
|  | $t = 10, \lambda = 50, r_0 = 2.2$ | 92.26 | 87.37 | 61.92 |
|  | $t = 0.1, \lambda = 1500, r_0 = 2.2$ | 11.35 | 11.35 | 11.35 |

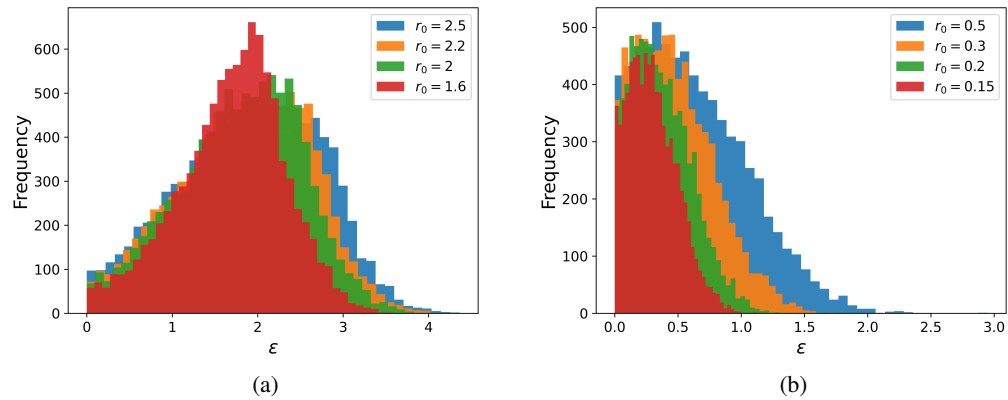

Figure 4: Comparison of the effect of the parameter $r_0$ on the certified radius of test data points for (a) MNIST and (b) CIFAR-10.

Table 6: Sensitivity to CRM hyperparameters on CIFAR-10 (6C2F)

|  | Hyperparameter setting | Clean (%) | PGD (%) | Certified (%) |
|---|---|---|---|---|
| Original setting | $t = 5, \lambda = 15, r_0 = 0.2$ | 74.82 | 72.31 | 64.16 |
| Effect of $\lambda$ | $t = 5, \lambda = 1, r_0 = 0.2$ | 83.19 | 73.45 | 0.0 |
|  | $t = 5, \lambda = 5, r_0 = 0.2$ | 82.87 | 73.57 | 0.0 |
|  | $t = 5, \lambda = 20, r_0 = 0.2$ | 72.0 | 69.67 | 62.65 |
|  | $t = 5, \lambda = 50, r_0 = 0.2$ | 67.32 | 64.94 | 59.68 |
| Effect of $r_0$ | $t = 5, \lambda = 15, r_0 = 0.15$ | 75.66 | 73.2 | 63.47 |
|  | $t = 5, \lambda = 15, r_0 = 0.3$ | 71.4 | 69.03 | 63.46 |
|  | $t = 5, \lambda = 15, r_0 = 0.5$ | 66.79 | 64.79 | 60.99 |
| Effect of $t$ | $t = 3, \lambda = 10, r_0 = 0.2$ | 83.64 | 74.26 | 0.0 |
|  | $t = 3, \lambda = 20, r_0 = 0.2$ | 68.38 | 65.95 | 60.68 |
|  | $t = 7, \lambda = 20, r_0 = 0.2$ | 74.03 | 71.54 | 63.07 |
|  | $t = 7, \lambda = 50, r_0 = 0.2$ | 69.87 | 67.43 | 61.15 |
|  | $t = 10, \lambda = 30, r_0 = 0.2$ | 74.08 | 71.61 | 62.92 |
|  | $t = 10, \lambda = 100, r_0 = 0.2$ | 68.49 | 66.04 | 59.64 |

Table 7: Sensitivity to CRM hyperparameters on Tiny-Imagenet (8C2F)

|  | Hyperparameter setting | Clean (%) | PGD (%) | Certified (%) |
|---|---|---|---|---|
| Original setting | $t = 12, \lambda = 150, r_0 = 0.2$ | 23.97 | 23.04 | 17.98 |
| Effect of $\lambda$ | $t = 12, \lambda = 100, r_0 = 0.2$ | 41.22 | 30.58 | 0.0 |
|  | $t = 12, \lambda = 130, r_0 = 0.2$ | 24.44 | 23.2 | 17.8 |
|  | $t = 12, \lambda = 170, r_0 = 0.2$ | 23.74 | 22.74 | 17.66 |
|  | $t = 12, \lambda = 200, r_0 = 0.2$ | 22.1 | 21.3 | 16.84 |
| Effect of $r_0$ | $t = 12, \lambda = 150, r_0 = 0.15$ | 25.34 | 23.82 | 18.14 |
|  | $t = 12, \lambda = 150, r_0 = 0.3$ | 22.54 | 21.38 | 17.14 |
| Effect of $t$ | $t = 5, \lambda = 110, r_0 = 0.2$ | 41.54 | 28.84 | 0.0 |
|  | $t = 5, \lambda = 140, r_0 = 0.2$ | 16.88 | 16.36 | 14.42 |
|  | $t = 10, \lambda = 150, r_0 = 0.2$ | 22.86 | 22.06 | 17.56 |
|  | $t = 10, \lambda = 180, r_0 = 0.2$ | 22.36 | 21.36 | 17.06 |
|  | $t = 15, \lambda = 170, r_0 = 0.2$ | 24.68 | 23.68 | 17.46 |
|  | $t = 15, \lambda = 200, r_0 = 0.2$ | 23.92 | 22.82 | 17.7 |

in the loss function, and instead of using $h(x, y; \theta, t) = \begin{cases} -\underline{R}_t^{\text{soft}}(x, y; \theta, t) & \underline{R}(x, y; \theta) \leq r_0 \\ 0 & \underline{R}(x, y; \theta) > r_0 \end{cases}$, we

use $h(x, y; \theta, t) = \begin{cases} \frac{1}{t} e^{-t\underline{R}(x,y;\theta)} & \underline{R}(x, y; \theta) \leq r_0 \\ 0 & \underline{R}(x, y; \theta) > r_0 \end{cases}$ that provided better experimental results for this scenario.

### B.1.3 Run Time Analysis

To support the arguments about time complexity in A.4, we analyze the run time of LipLT and its use in the CRM training procedure. For this purpose, we use the `torch.cuda.Event` functionality to accurately time events. We perform the training using either CRM or standard loss functions and with either the naive Lipschitz bound or LipLT. For MNIST and CIFAR-10 we use a batch size of 512 and a batch size of 256 for Tiny-Imagenet. Unless otherwise stated, we use 5 iterations of the power method for the Lipschitz calculations.

**Model Depth.** We first study the effect of the depth of the model on the Lipschitz calculations. For this, we start with the 6C2F model of CIFAR-10 and increase the number of layers. To save space, we note that all models have the same first 6 convolutional layers. We thus let `6C` represent the first 6 convolutional layers that appear in the 6C2F architecture. Furthermore, let `C1 = C(64, 3, 1, 1)` and `C2 = C(64, 4, 2, 1)`. The tested models are then as follows:

- 7C2F: `6C, C1, L(512), L(10)`
- 8C2F: `6C, C1, C2, L(256), L(10)`
- 9C2F: `6C, C1, C2, C1, L(256), L(10)`
- 9C3F: `6C, C1, C2, C1, L(128), L(128), L(10)`
- 9C4F: `6C, C1, C2, C1, L(128), L(128), L(64), L(10)`
- 10C4F: `6C, C1, C2, C1, C2, L(64), L(64), L(32), L(10)`
- 11C4F: `6C, C1, C2, C1, C2, C1, L(64), L(64), L(32), L(10)`

Figure 5 displays the linear trend that we claimed, thanks to the parallel implementation.

**Number of power iterations** We study the effect of the number of loops for the power method on the run time of the algorithm. To do so, using the corresponding model for each dataset from the original experiments, we clock the time for the calculation of the Lipschitz constant and a full iteration of training using the CRM loss with both the naive Lipschitz bound and LipLT. The experiments use pairwise Lipschitz calculations with the same approximate setting for the larger case of the 8C2F model explained in appendix B.1.1. Figure 6 portrays the results of the experiments. As per the complexity analysis conducted in A.4, the effect of the number of power iterations is linear.
To further study the Lipschitz calculation process itself, we consider it in isolation and outside the training loop in Figures 6e and 6f. To do so, we turn off the gradient calculation[18] and only calculate

---

[18] We use the `torch.no_grad()` context manager in `Python`.

Table 8: Analysis of the effect of the different components of CRM on the accuracy of the trained model.

|  | Soft Radius | | | Hard Radius | | | Naive Lipschitz | | |
|---|---|---|---|---|---|---|---|---|---|
|  | Clean | PGD | Certified | Clean | PGD | Certified | Clean | PGD | Certified |
| Pairwise | 74.82 | 72.31 | 64.16 | 78.28 | 75.46 | 62.62 | 62.29 | 60.24 | 54.68 |
| Per class | 64.74 | 62.85 | 56.47 | 77.38 | 74.88 | 59.27 | 58.47 | 56.47 | 50.33 |
| Per network | 74.00 | 71.27 | 63.19 | 77.56 | 74.54 | 61.84 | 61.6 | 59.63 | 53.81 |

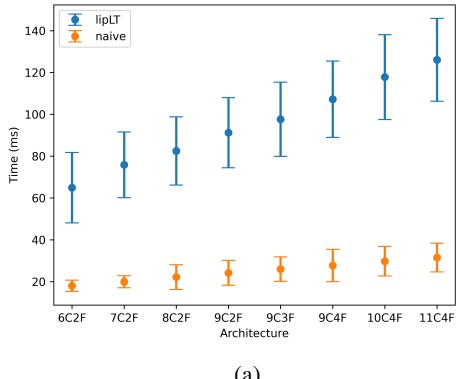 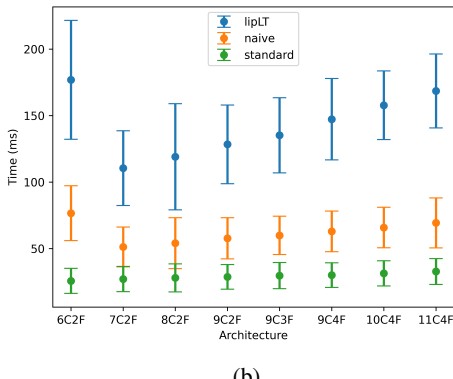

(a)                                                          (b)

Figure 5: GPU implementation of LipLT depicts the claimed linear time complexity. (a) Time spent on the calculation of pairwise Lipschitz constants. (b) Time for a full iteration of training.

the Lipschitz constant for the given model for many iterations. These figures portray a much more consistent linear growth.

**Dataset** The complexity analysis provided in A.4 is in terms of the number of passes through different weights of the network, but not in terms of the actual input size. To also provide a study in this regard, we run experiments comparing the effect of the input size. To do so, we fix a general network architecture for all three datasets and only change the number of neurons in the fully connected layers to fit each dataset. We use the 6C2F model of the CIFAR-10 dataset as basis. Figure 7 provides the results of the experiments.

### B.1.4 Evolution of Lipschitz Constants

We study the evolution of the average pairwise Lipschitz constants for CIFAR-10 under different training scenarios. Figure 8 provides the results. As evident from the figure, the average pairwise Lipschitz constant grows during the first few iterations as the training includes a warm-up phase, after which the regularizer is added to the loss function. The effect of the learning rate adjustment starting from epoch 200 is also present in all figures, with a larger impact on the standard training. Furthermore, when using the naive Lipschitz estimation for training, we can see that the gap between the naive Lipschitz estimation and LipLT is very small (needless to say that LipLT still provides a smaller Lipschitz constant). We note that in this scenario LipLT does not provide a significant improvement over the naive Lipschitz constants. However, referring to Table 8, using the naive Lipschitz estimate hinders the expressivity of the network and hurts the clean accuracy.

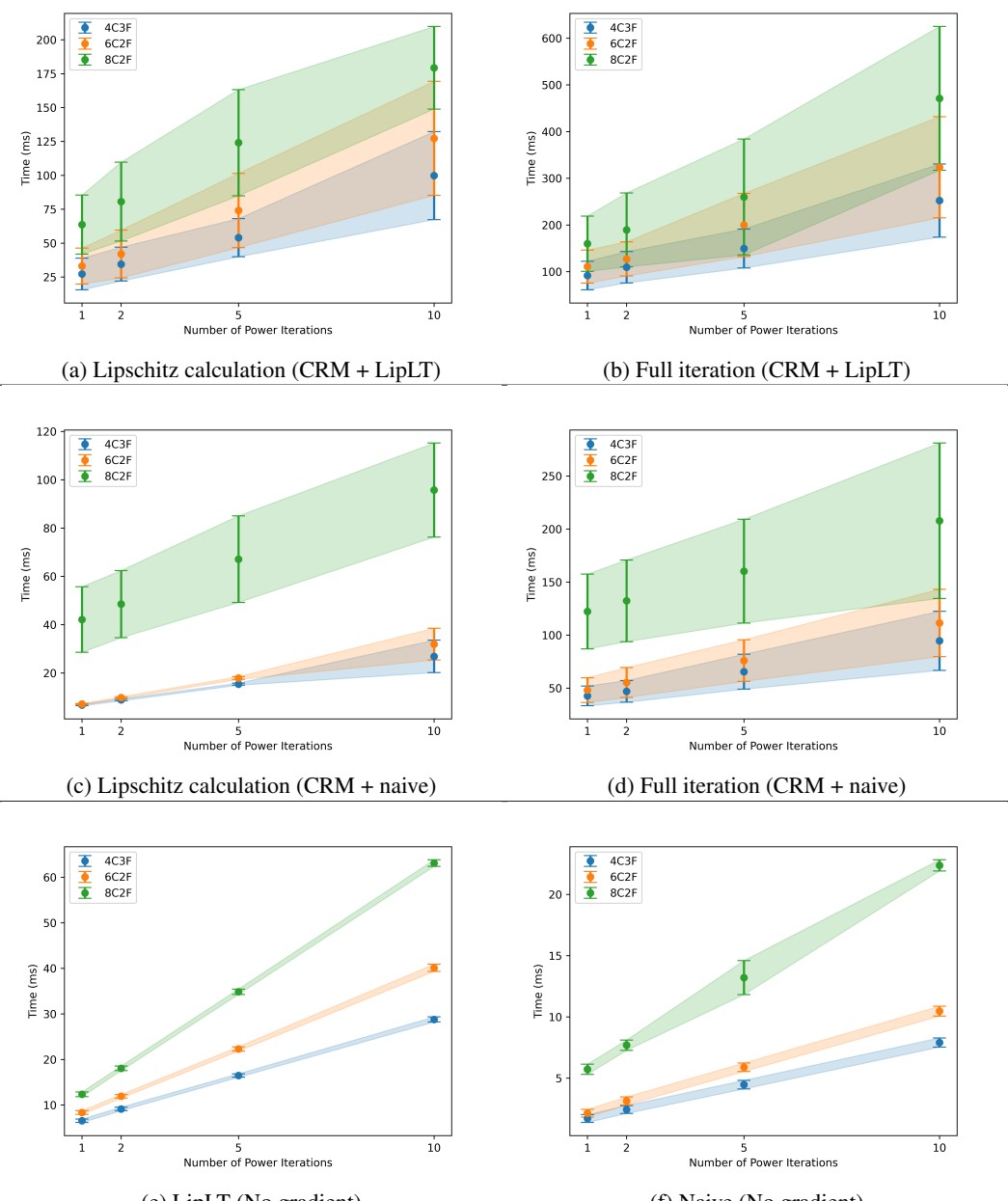

(a) Lipschitz calculation (CRM + LipLT)

(b) Full iteration (CRM + LipLT)

(c) Lipschitz calculation (CRM + naive)

(d) Full iteration (CRM + naive)

(e) LipLT (No gradient)

(f) Naive (No gradient)

Figure 6: Effect of the number of power iterations on the run times of the Lipschitz calculation and the training process for the different datasets. (a, b) CRM + LipLT, (c, d) CRM + Naive. (a, c) Run times for calculating the Lipschitz constants during training. (b, d) Run times for one full training iteration, involving forward propagation, Lipschitz calculation, backward propagation, and weight update. (e/f) Pure LipLT/naive Lipschitz estimation without any gradient calculations In (e, f) we only evaluate the Lipschitz estimation algorithm outside of the training process and without any gradient flow.

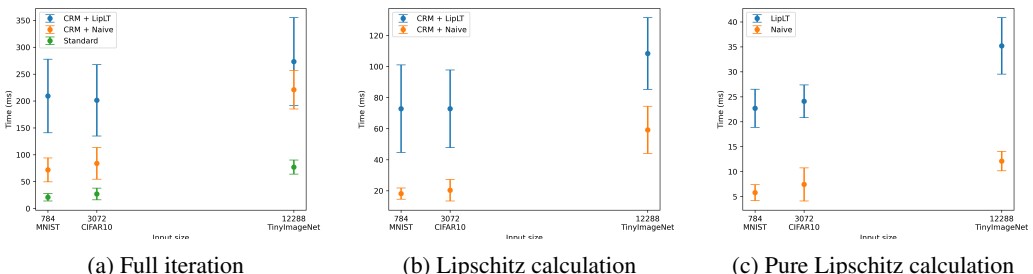

(a) Full iteration          (b) Lipschitz calculation          (c) Pure Lipschitz calculation

Figure 7: Effect of the input size on the run time of the algorithm. (a) Run time of one full iteration of training. (b) Run time of Lipschitz calculation during training. (c) Pure Lipschitz calculation (no gradient flow) run time.

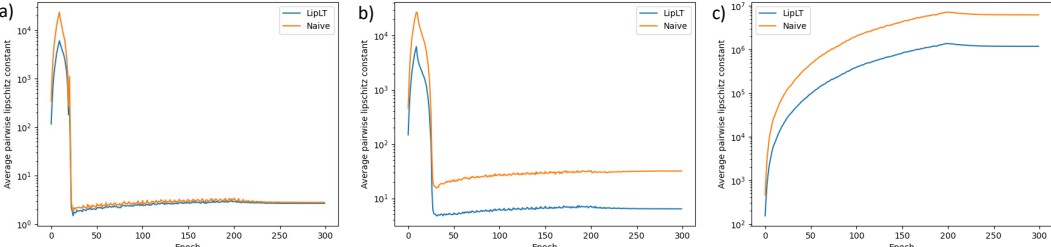

Figure 8: Evolution of the Lipschitz constant over training under different training settings. We calculate the average pairwise Lipschitz constant using both the LipLT and naive Lipschitz estimation methods. (a) CRM + Naive. (b) CRM + LipLT. (c) Standard training.