# OpenReview forum: "Certified Robustness via Dynamic Margin Maximization and Improved Lipschitz Regularization"
_NeurIPS.cc/2023/Conference — NeurIPS 2023 poster_

### Official Review · Reviewer_XLJR · 2023-07-05

**Soundness:** 4 excellent
**Presentation:** 3 good
**Contribution:** 2 fair
**Rating:** 5
**Confidence:** 4

**Summary:**

This paper proposes a new bound for margin maximization, as well as tighter estimation of Lipschitz constant, to improve the adversarial robustness of neural networks. Experiments on three benchmark datasets are presented to show the proposed method can improve the certified radius.

**Strengths:**

- This paper is written well and clearly
- New bound for input margin as regularization term
- New estimation of Lipschitz constant for multi-layer networks
- Experiments are conducted to show improved robust radius

**Weaknesses:**

- Regarding the improved bound for input margin, I think the contribution is limited since the similar strategy has be used frequently in recently works.
- The authors argued that the estimation of Lip. constant is scalable for multi-layer networks, but experimental results are not verified.

**Questions:**

See above.

**Limitations:**

The novelty of margin regularization is limited and the experimental results are not sufficient.

---

> ### Author Rebuttal · Authors · 2023-08-10
>
> We would like to thank the reviewer for the comments.
>
> >Regarding the improved bound for input margin, I think the contribution is limited since the similar strategy has be used frequently in recently works.
>
> > The novelty of margin regularization is limited and the experimental results are not sufficient.
>
> Kindly refer to our global response, where we highlight the novel aspects of the proposed regularizer, and how it combines the advantages of various proposed methods in the literature. With the added experiments in the rebuttal (please see the global response and the attached pdf), we believe that our theoretical results are now backed by experiments.
>
> > The authors argued that the estimation of Lip. constant is scalable for multi-layer networks, but experimental results are not verified.
>
> Please see our global response “Complexity Analysis of Lipschitz Estimation Algorithm”  in which we provide a complexity analysis of our Lipschitz estimation algorithm. We have done more experiments to show the scalability of our Lipshschitz estimation method. In Figure 4 of the attached pdf, as we increase the number of layers, the Lip. calculation runtime increases almost linearly, similar to the naive bound.

---

> > ### Comment · Reviewer_XLJR · 2023-08-17
> > **Thank for the rebuttal**
> >
> > Thank you for the authors' rebuttal regarding more experimental results and explaination on the novelity.

---

### Official Review · Reviewer_4zXU · 2023-07-05

**Soundness:** 3 good
**Presentation:** 3 good
**Contribution:** 3 good
**Rating:** 7
**Confidence:** 3

**Summary:**

The authors explore how a model’s robustness to local pertubations in the input space can be enhanced through regularization at a low cost on the clean accuracy.
The goal is to have large margins on the desicion boundaries in input space.
Since adding a regularization directly on the size of these margins is very costly, the authors introduce an alternative and efficient proxy to regularize this.
They use the Lipschitz constants of the logit difference function between pairwise classes to identify the most important directions to regularize.
Since computing the constants is exactly is costly, but naive estiamtes are very conservative as they do not take into account the correlation between the trained layers, the authors introduce a new approximation that is efficient to compute.
They show that the method is comparing well with SOTA adversarial robustness methods.


**Strengths:**

1. The authors explore a new regularizer that is efficient to compute and can trade off the margin maximation in the input space via Lipschitz regularization which is still efficient to be used on classical toy datasets. They show that this regularizer is on par with existing approaches and may even outperform them.
2. They provide a new strategy to efficiently approximate Lipschitz constants which is of independent interest.
3. The paper overall is well written and structured; and can be followed comfortably.


**Weaknesses:**

1. In the experimental evaluation, I would find it valuable to study your model also independently to see how the accuracy of the improved Lipschitz estimate affects the final accuracy outcome. The claim that the less conservative estimates lead to a better regularizer can be checked independently of other state-of-the-art methods. I think interesting experiments to support that claim could be: Comparing how the estimated (and naive) Lipschitz constant evolves throughout training for training with and without regularizer for the same architecture; replacing the improved estimate of the Lipschitz constant with the naive one; introducing a global hyperparameter that replaces L_{ij} and tunes it at the computational budget available to the Lipschitz constant estimation. These are just a few suggestions that would improve the understanding of the actual efficiency of the introduced loss regularizer.
2. The method relies on estimates for Lipschitz constants for every pair of classes, this scaling becomes unreasonable for datasets with many different classes.

**Questions:**

1. In L.3 you mention Lipschitz-capped networks and argue on to say that these approaches are not good at increasing the input margin. Why is that exactly, does your own approach not follow a similar idea?
2. L.90, can you elaborate in a half-sentence on the problems of applying this to multilayer networks?
3. Can you comment at some point, on why you exclude wrongly classified examples from the regularizer and if this sequential inclusion is expected to have some effect?
4. I like the notation with the underline for the differentiable functions. However, since \underline{R}_t^soft is not computing the same version as the R_t^soft and is more closely related to \underline{R} I find the repetition of the names not optimal. Especially in L.249-252 the notation is confusing. First, in L. 251 there is a repetition (replace … with … with) and also it is not clear to me if h(x) in L.249 takes x=R_t^soft(x’,y,theta) so that the condition is a double application of R_t(R_t^soft(…)). I tried to understand it in different ways but I settled on the following to interpret the results: replace R(..) by x only in the case distinction in L.250 and the second ‘with' in L.51 to ‘and’. Could you please clarify if that is correct?
5. It would be nice if you could spend 2 more sentences explaining what you mean by loop transformation. To me the term was new and difficult to google, I checked the appendix to understand what was going on.
6. In (17) you use the function psi(.) which is not introduced in the main draft, only in the appendix - maybe you can explain that jointly with the loop transformation.
7. I did not find a definition of the Standard baseline, so I am just guessing that it is normal training without any regularization…
8. Can you give an intuition why you chose the specific values of l_2 perturbations in L.231/232? Is there a good reason why this was not tested for other values?
9. Can you give a clearer comparison how costly the computation of the approximate Lipschitz constants is in comparison to the backpropagation in terms of GPU time in your expeiments? O(L) is mentioned but it would be nice to understand what this means in practice.


Minor comments:
1. In L.81 introduce the abbreviation SDP
2. L.88 constantS
3. L.95 To keep with convention, iterate over small i instead of small k in the CE
4. L.158 eqref missing
5. In (10) you just say ‘’compatible dimensions’ but then use dimensions in Thm.3.1 without introduction  — it would be nice to make them clear from the start if you are going to use them anyways.
6. L.185 Abbreviation of LMI not introduced
7. L.189 blockS
8. The numbering of the equations in the Supplementary material is very confusing, as theorems and text have different numberings but are not distinguished (so the eq numbers are somehow in seemingly random order)

**Limitations:**

The authors are clear about the limitations of their method in the paper. I would like to add that it is not immediately clear to me how more complex architectures would be amenable to similar estimates of the Lipschitz constants. Adversarial robust training which only relies on adapting the training data can achieve this for any architecture, so if this is not given, I would include it as a limitation.

---

> ### Author Rebuttal · Authors · 2023-08-10
>
> We thank the reviewer for their perceptive remarks and suggestions. We will add your useful experiment and ablation study suggestions to the final manuscript. We'll try to address your concerns in the following:
>
> ### Addressing weaknesses
> - In the limited rebuttal period we were able to incorporate two of your suggestions. First, we trained three networks under different training methods and logged the average pairwise Lipschitz constants using both the naive and improved methods after each epoch of training. The training methods are as follows (CRM represents our proposed loss function):
>   - CRM with improved Lipschitz estimates at train time
>   - CRM with naive Lipschitz estimates at train time
>   - Standard clean training
>
>   The results of these experiments are provided in Figures 1, 2, and 3 of the attached pdf, respectively. To better showcase the data we have used a logarithmic scale for the y-axis.
>
>   We also performed a set of runs to study the effect of each component. The results are reported in Table 1 of the attached pdf. We performed the analysis on CIFAR10. We used two sets of hyperparameters that yielded good results from our experiments under different scenarios and reported the best number in Table 1. The different columns are as follows:
>   - Soft Radius: Training using LogSumExp (with our improved Lipschitz).
>   - Hard Radius: Training using
>   $  \underline{R} = \min_{i \neq y} \frac{z_y - z_i}{L_{yi}} $
>   (with our improved Lipschitz).
>   - Naive: Training using LSE but with naive Lipschitz estimation.
>
> - **For a moderate number of classes**: computing pair-wise Lipschitz constants would only require changing the last linear layer. This means that a lot of operations in our algorithm would remain the same, which we exploit. Kindly see Appendix A.3.2 for more details.
>
>   **For a large number of classes**: we can use two levels of relaxation:
>
>   - $L_{ij} \leq L_i + L_j$: this will reduce the number of Lip. calculations from quadratic (in # of classes) to linear.
>
>   - $L_{ij} \leq \sqrt{2} L$: this will reduce the number Lip. calculations from quadratic to constant.
>
>   Nevertheless, regardless of what relaxation we use, our Lipschitz estimation algorithm provably improves the naive bound while being efficient.
>
>
> ### Questions
> - Q1: Lipschitz-capped neural networks put a pre-defined bound on the Lipschitz constant, which would guarantee a lower bound on the certified radius. However, this lower bound can be loose and hence, ineffective (please see paragraph #618). By directly regularizing a tighter lower bound on the certified radius, we will have a more direct control over the certified radius
> - Q2: For multi-layer networks, the resulting LMI will be tri-block-diagonal. It is not possible to make the LMI block diagonal using the proposed technique in [39]. The same reference has left the multi-layer case for future work.
> - Q3: If an example is classified wrongly, increasing its margin would make it **robustly wrong**, i.e., away from the classification boundary but in the wrong class. To avoid this, we should regularize points only when they get classified correctly during the training process. Since initially, many points are classified wrongly, we start the robust training after a few epochs of normal training (i.e., w/o the regularizer), so that a relatively good proportion of points will be classified correctly.
> - Q4: We apologize for the confusion. In the revised manuscript, we are now using the same $h$ across all data sets. Specifically, we are now using  $h(x, y; \theta) = \begin{cases}  -\underline{R}_t^{soft}(x, y; \theta) & \underline{R}_t(x, y; \theta) \leq r_0 \\\\ 0 &  O.W. \end{cases} $. This $h$ was only used for the MNIST dataset, but now we are able to obtain similar performance on other datasets using the same function $h$.
> - Q5: Kindly refer to our global response for a detailed explanation of this.
> - Q6: Thanks for the great suggestion. We intend to bring the $\psi$ notation to the main body to better explain the concept of loop transformation. This would require condensing Section 2 to open up some space.
> - Q7: Yes you are correct. The standard baseline is for when the network is trained using only the cross entropy loss function.
> - Q8: These are the $\ell_2$ norm bounds that are common in previous work [19, 24]. We would like to point out that Fig. 1 of the paper does portray the distribution of the certified radii.
> - Q9: We kindly refer the reviewer to the section “Complexity Analysis of Lipschitz Estimation Algorithm” in the global rebuttal for an overview of your comment.  Furthermore, we present the following table addressing your comment.
>
>   We used the *torch.cuda.Event* functionality to precisely time our algorithm. The experiments were conducted on a single NVIDIA RTX A5000 with 24GB of RAM. We used a batch size of 512 for training CIFAR10. The numbers in the table show the average time $ \pm $ standard deviation in milliseconds (ms). CRM is short for our training method. LipLT is our improved Lipschitz estimation algorithm. In the table, *Lipschitz calculation* measures only the time required to calculate the Lipschitz constant using the relevant method in each iteration, and *Full iteration* measures the full time required for that particular iteration.
>
>   |   | CRM with LipLT | CRM with Naive | Standard|
>   |--------|--------|--------|--------|
>   | Full iteration | 122.7 $\pm$ 27.2 | 51.6 $\pm$ 10.9 | 16.4 $\pm$ 7.1 |
>   |Lipschitz calculation | 43.1 $\pm$ 6.5 | 13.3 $\pm$ 1.1 | $\quad-$ |

---

> > ### Comment · Reviewer_4zXU · 2023-08-14
> >
> > I would like to thank the authors for their careful consideration and additional experiments as an answer to all reviews which enhance the presentation of their idea. I think the paper is intersting and nicely done.

---

### Official Review · Reviewer_9vGY · 2023-07-05

**Soundness:** 3 good
**Presentation:** 2 fair
**Contribution:** 3 good
**Rating:** 6
**Confidence:** 4

**Summary:**

This paper developed a robust training algorithm whose objective is to increase the margin while regularizing the Lipschitz constant of the model along vulnerable directions. The authors also developed an improved Lipschitz estimation for a multi-layer networks with residual connections. Experiments on MNIST and CIFAR-10 shows superior performance of the proposed method compared to the state-of-the-art.

**Strengths:**

- A novel regularized loss function that directly promote larger margins in the input space is proposed.
- A scalable Lipschitz bounding algorithm is proposed, which is tighter than the naive bound while being efficient.
- Experiments on MNIST and CIFAR-10 showed the effectiveness of the proposed methods.

**Weaknesses:**

- This paper proposed many interesting methods, but I sometimes find it hard to see the motivation of each proposed method and the connections between them. It may be good to spend some more space to set up the preliminaries:
    - The name is section 2 is a unified perspective on margin maximization, I'd hope to see what margin maximization algorithms it unified.  For instance, the variant of (6) may result in different algorithms, replacing L_{yi} with \sqrt(2)L will results to Lipschitz margin training [Tsuzuku et al. 2018].
    - It would be good to decouple the effectiveness of each component: the use of pairwise Lipschitz, the use of log-sum-exp to make max/min to be differentiable, the use of the new Lipschitz bounding algorithm. It would be interesting to see the contribution from each of these to the certified accuracy.

- It would be good if the authors can discuss the sensitivity of the proposed method wrt the choice of hyper-parameters. An ablation study on CIFAR-10 with various hyper-parameters and the corresponding certified robustness performance would be helpful. The proposed method is not performing as well on Tiny-Imagenet, I understand that it may be because of the inferior choice of hyper-parameters, but it may be good to debug more into that, especially giving table 2, the improved pair-wise Lipchitz of Tiny Imagenet seems to be much better than other Lipschitz, it is surprising that it does not lead to better certified robustness.

- It would be nice to provide the computational cost of the proposed method in terms of Flops or running time.

**Questions:**

- In section 3, the proposed scalable Lipschitz estimation method seems to be work on neural networks with residual connections (Eq 16). But in the experiments, the model architectures seems to not have residual connections. How to apply the proposed bounds then?

- What is loop transformation? I am not familiar with the term.

- Line 221, Recurring structure can reduce the complexity by dynamic programing, but the operations are still sequential rather than parallel since we need m_j, j<=k to compute m_{k+1}. How does the parallelized implementation on GPU work?


**Limitations:**

Yes

---

> ### Author Rebuttal · Authors · 2023-08-10
>
> We thank the reviewer for their thoughtful suggestions and constructive criticism. In the following, we respond to the weaknesses raised by the reviewer.
>
> ### Addressing weaknesses
> - Our hope was to present a unifying loss function in Eq. (6) that can recover several existing loss functions. For example, in Line #618, we showed that regularizing the Lip. the constant of the whole network is simply an upper bound to our regularizer. Overall, we tend to agree that this may not be the best title for the section, and we propose to change the title to “Dynamic margin maximization for neural network classifiers” if the paper gets accepted.
> - We have run new experiments to further portray the effect of the different parts of the algorithm as you had kindly suggested. The results are reported in Table 1 of the supplementary pdf of the global rebuttal. We performed the analysis on training CIFAR10. We used two sets of hyperparameters that yielded good results from our experiments under different scenarios and reported the best number in Table 1. The different columns are as follows:
>   - Soft Radius: Training using $ \underline{R}^{soft}$ (Eq. (7)) + our Lipschitz estimation algorithm.
>   - Hard Radius: Training using $ \underline{R} = \min_{i \neq y} \frac{z_y - z_i}{L_{yi}} $ (Eq. (6)) + our Lipschitz estimation algorithm.
>   - Naive: Training using $ \underline{R}^{soft}$ (Eq. (7)) + naive Lipschitz estimation.
>
> - In addition to Table 1 provided in the attached pdf, we would like to refer the reviewer to tables 5, 6, and 7 in the appendix of the original submission that perform several sensitivity studies.
>
> - In our revised manuscript, we have addressed this comment and have added a section on the complexity analysis of our Lipschitz estimation algorithm. For a brief summary of our analysis, you can refer to the global rebuttal section "Complexity Analysis of Lipschitz Estimation Algorithm." We have portrayed the claimed Linear performance in Figure 4 of the supplementary pdf. Furthermore, we have provided run times in the following table for your consideration. We used the *torch.cuda.Event* functionality to precisely time our algorithm. The experiments were conducted on a single NVIDIA RTX A5000 with 24GB of RAM. We used a batch size of 512 for training CIFAR10. The numbers in the table show the average time $ \pm $ standard deviation in milliseconds (ms). CRM is short for our training method. LipLT is our improved Lipschitz estimation algorithm. In the table, *Lipschitz calculation* measures only the time required to calculate the Lipschitz constant using the relevant method in each iteration, and *Full iteration* measures the full time required for that particular iteration.
>
>   |   | CRM with LipLT | CRM with Naive | Standard|
>   |--------|--------|--------|--------|
>   | Full iteration | 122.7 $\pm$ 27.2 | 51.6 $\pm$ 10.9 | 16.4 $\pm$ 7.1 |
>   |Lipschitz calculation | 43.1 $\pm$ 6.5 | 13.3 $\pm$ 1.1 | $\quad-$ |
>
>  ### Question
> - Q1: We initially based our architecture as that of equation (12), derived from [17], to provide further insight on the previous works that use this architecture. To maintain generality, we provided our Lipschitz estimation algorithm for this more general architecture. To provide a more fair comparison with relevant previous work such as [19, 24, 18], we used feed-forward neural networks without residual connections. As our results in Table 1 of the main manuscript portray, we are able to acquire similar or better results than other more complicated residual networks that use many more layers. Thus, we only conducted our experiments on normal feed-forward networks. In case your question also regards the applicability of our derived Lipschitz estimation algorithm to feed-forward networks, we note that this is acquired by setting $H=0$ and $G=I$. In our appendix section A.3, from equation (35) onwards, we derive the relevant formulas for normal feed-forward networks. Furthermore, in A.3.2 we describe all the implementation notes that allow our method to run in Linear time. We kindly ask the reviewer to further elaborate on their question if these comments do not address their concern.
>
> - Q2: Please refer to the global rebuttal section "Details on Loop Transformation."
>
> - Q3: Your observation of the sequential structure required for calculating $m_{k + 1}$ is valid. However, we emphasize that the main point of complexity for calculating $m_{k + 1}$ is calculating the different matrix norms that appear. The way we prevent this sequential structure from slowing down our computation is that our algorithms first calculate all the different norms that appear in all the terms of $m_i$ for $ i \leq L$. After these norms have been calculated, we are able to calculate the $m_i$ sequentially as they consist of only scalar-scalar multiplications. The details of the algorithm are involved and we refer the reviewer to algorithm 2 of our appendix.

---

> > ### Comment · Reviewer_9vGY · 2023-08-17
> >
> > Thank the authors for their detailed reply. I am ok with accepting the paper.

---

### Official Review · Reviewer_U11m · 2023-07-07

**Soundness:** 2 fair
**Presentation:** 2 fair
**Contribution:** 2 fair
**Rating:** 5
**Confidence:** 4

**Summary:**


This paper proposes a loss to induce robustness w.r.t. adversarial attacks. This loss is composed of a term for enlarging the logit difference (such as classical  Cross-Entropy Loss), and a surrogate term to  maximize the robust radius. The latter imply to compute an upper bound of the Lipschitz constant (called L_ij) of the difference of logits z_y-z_i (y is the true class and any other class i).
Inspired by work in [39], the authors propose to take into consideration the motonicity of activation functions to compute a tigher lower bound of a residual-like layer than the classical one. They also propose an extension for computing an upper bound of the Lipschitz constant of a multilayer residual NN.
Note that Batchnorm layers and dropout are not supported.
They also claim to compute these L_ij with the same complexity as computing the Lipschitz constant of each layer (supplementary materials)
Experiments are done on MNIST,CIFAR10 and TinyImagenet. Comparison is provided with several methods, reporting clean accuracy, adversarial accuracy (PGD attack) and certified accuracy, and report higher certified accuracy. They also compare the Lipschitz estimation to classical bounds (sqrt(2)Lz, Li+Lj and theirs Lij) and claims an improvement of up to 200.


**Strengths:**

The algorithm described in section 3.1 seems new and results presented in table 1 indicate that it enhance the lipschitz estimation..

**Weaknesses:**

-Clean performances are low and equivalent to 1LipchitzNN but author indicate in the related work that it should have a higher expressivity

-No evaluation of computational complexity is provided. Scalability os the method is not clear


**Questions:**

The paper is rather easy to read. Even if without the supplementary materials,
Several errors or inexactitude can be found in  Notation and equation and proof
Computation of a tight upper bound of NN seems new and interesting. However enlarging the margin under lipschitz constraint or regularization is not new. And besides main loss and regularization may have contradictory objectives (see detail comments)

Detailed remarks:

L31: correlation between robustness and lipschitz constant of NN is an established property , cite[1]
L33: “constraining the model (e.g. 1-Lipschitz models) can lead to excessive regularization in addition to potentially limiting expressivity.”: it has been shown that 1-Lipschitz NN have the the same expressivity has non constrained networks (see “pay attention to your loss”, Bethune et al. Neurips’22). Thus this sentence is false.
L43: “hinges on the idea of loop transformation on the nonlinearities”: the notion of “loop transformation” is not clear. Please clarify or add a reference.
L54: choice of tiny-imagenet is not justified. Authors claim for scalability (L84) but restrain the learning to tiny-Imagenet.This must be explained
L92: notation for matrix norm  is not described in this section, and ||.|| can be confusing: frobenius nor, spectral norm,…
L100: “A unified perspective on margin maximization” : this section title is not justified. Please explain which unification is provided in this section?
L111: “we note that maximizing the logit margin…..in the input space”: this is a well known property ([1])
L125: “many choices for h, we use the same function proposed in [17] (see supplementary materials) “ but no information is given in the supplementary matrials
L132: eq 4 this equation is not use in the following, confusing in notation with eq (7), and not evaluated. Probably useless for paper understanding
L134: .”Solving (4) still requires an iterative inner loop within the outer loop of training”: transformation from inf formulation to outer loop is unclear. Please precise if this loop is an adversarial search or any other method
L146: “This lower bound is not differentiable”: min function is differentiable. A least subderivative. Justification l149 is a better one.
Eq 9: it is known that CE loss aims to increase the Lipschitz constant of the NN during training (see paper Bethune et al cited earlier), and the regularization in (6) aims to reduce this Lipschitz constant. Please explain how these two antagonist objectives will be balanced.
L169: inequality soes not represent the lispchitz constraint. A divide operator is missing
Theorem 3.1: this theorem isbased on supposition of existence of \rho and T, but in the following justifications of this existence is not clear
L185: “analytic solutions to the LMI of Theorem 3.1,”: acronym LMI is not defined
L192: same remark as on line L33
L204: ||H|| isnot clear can be ether frobenius norm or spectral norm?
L247: “,we calculate the certified radii using(6)” (6) is only applicable for ReLU like activation. Please justify how this certified robustness is computed for NN such as MaxMin (2)
L249: “CRM hyperparameters”: first use of this acronym. Please detail
L251: contradiction of justification given in eq (8). Please justify why the choice of h is not the same for each dataset.
L255: “For Tiny-Imagenet, as the number of classes is far larger, bounding the pairwise Lipschitz would be computationally heavy.” Such a sentence and footnote 4  need justification. Please detail the computational complexity induced by your method
Table 1: In related works, author indicate that 1-Lipschitz NN are harder to train and have a lower expressivity. However, clean performance of the proposed solution has performances equivalent to   1Lip solutions and far from classical performances. Please justify
No software provided to verify the correctness of the paper


Supplementary materials:
L532: “However, the expressive power of 1-Lipschitz networks needs to be addressed and some works study their limitations [47]. ” this sentence is false. See remark on line 33
L580: \gamma is non differentiable. : Max operation is differentiable?
L598: h function was supposed to be described in the supplementary materials
L605 (equation) replace (1-t) by (1-t^hat)
L615 (equation) i\neqy is replaced in this eq by i\in[1,K-1] pleae justify?
L638: please justify te existenceof T and \rho?
L660 m_k is not the lipschitz constant, but an upper bound of the Lipschitz constant
L660 (eq) m_{k+1} should be lower than
L673: eq 33 : lack ||
L719: “   Calculating the ℓ 2 norm of a general rectangular matrix can be performed by
using the power iteration”: confusing notations: power iteration is for spectral norm and is often in general notes ||.||_2 and not l_2. Confusing with frobenius norm

references: clean references for papers that are published and not preprint

**Limitations:**

Limitations are addressed

---

> ### Author Rebuttal · Authors · 2023-08-09
>
> We would like to thank the reviewer for their constructive criticism and detailed comments. We have provided a global response to address the comments that are common to all the reviewers. Here we address the comments specific to the reviewer.
>
> > Batchnorm layers and dropout are not supported.
>
> As batch normalization layers are linear, our method readily applies to them. In fact, if the batch normalization layer is next to a linear layer, the batch normalization could essentially be absorbed into the linear layer. The exact formula for the scenario in which batch normalizations are present in the architecture was not provided as it is architecture dependent. For future work, we hope to make the framework modular so that it can parse the architecture and adapt the algorithm.
>
> > Clean performances are low and equivalent to 1LipchitzNN...
>
> We are obtaining similar or better clean performance compared to 1-LipschitzNNs but with *smaller* architectures (please see Table 1 in the original submission)
>
> > No evaluation of computational complexity is provided. Scalability of the...
>
> We kindly refer the reviewer to the global response: “Complexity Analysis of Lipschitz Estimation Algorithm”, in which we provide a complexity analysis and support the analysis with new experiments. Remarkably, these experiments confirm the scalability of the method as the depth of the model increases.
>
> ### Detailed Remarks:
>
> -L33: We thank the reviewer for bringing the paper “Pay attention to your loss” to our attention, which had slipped through our literature review. We have removed the comments about the lack of expressivity of 1-Lipschitz neural networks.
>
> -L43: We agree with the reviewer. Please see the global response: “Details on Loop Transformation”, where we explain the details and provide a reference.
>
> -L54: The bottleneck in scalability is due to the use of pairwise Lipschitz constants and *not* the Lipschitz constant estimation algorithm itself. To elaborate, the number of pairwise Lipschitz constant calculations is $ k_m \choose 2$, where $k_m$ is the number of classes in the mini-batch. For datasets with a relatively small number of classes, it is very likely that each mini-batch chosen from the randomly shuffled dataset will contain samples from all classes. A workaround that we will investigate for larger datasets in the future is to modify the data loaders such that each mini-batch would only contain samples from a handful of classes: we would choose $k$ (relatively small) random classes, and draw random samples from these $k$ classes. We understand that this sampling might affect the performance of the model. Thus, we will leave a rigorous treatment of this approach to future work.
>
> Another possibility to improve the scalability for larger datasets is to use per-class Lipschitz bounds or the Lipschitz constant of the whole network.
>
> -L100: Our hope was to present a unifying loss function in Eq. (6) that can recover several existing loss functions. For example, in Line #618, we showed that regularizing the Lip. the constant of the whole network is simply an upper bound to our regularizer.  Overall, we tend to agree that this may not be the best title for the section, and we propose to change the title to “Dynamic margin maximization for neural network classifiers” if the paper gets accepted.
>
> -L125 In the experiments, we elaborated on what $h$ function we used. Hence, there was no need to refer to supplementary materials in line #125. We apologize for this. We have a paragraph in the supplementary material about $h$ (lines #594-599)
>
> -L134: For each data point (x, y), solving (4) requires an iterative optimization algorithm (the inner loop). After solving (4) for the whole minibatch, the algorithm then needs to update the weights of the model (the outer loop).
>
> -Eq9: There is an inherent trade-off between clean accuracy and adversarial robustness, as established in, for example, [17]. Our regularizer is used to balance this trade-off: by increasing the regularization parameter $\lambda$, adversarial robustness will tend to increase at the expense of reduced clean accuracy.
>
> -Thm3.1: Thanks for pointing out this subtle point. For simplicity, let’s assume $\alpha=0$. Suppose $T \succeq 0$ is chosen such that ${W^1}^\top W^1 - 2T \prec 0$. This is always possible. For example, choose $T = 0.5 (\sigma^2_{\max}(W^1)+\epsilon^2) I $.
>
> Then we can use Schur Complements for positive semidefinite matrices to obtain the following equivalent condition to the LMI:
>
> $-\beta^2 {W^0}^\top T ({W^1}^\top W^1 - 2T)^{-1} T W^0 \preceq \rho I$.
>
> Again, this is feasible by choosing $\rho$ to be at least equal to the maximum singular value of the positive definite matrix on the left hand side. In conlcusion, the LMI is always feasible. A similar analysis can be done for a more general case of $\alpha>0$. We will discuss the feasibility of this LMI in the appendix.
>
> -L247: Eq. (6) is applicable to MaxMin activations, but our Lipchitz constant estimation algorithm does not currently support such activations. We envision that we can handle GroupSort activations by representing them equivalently as multiple ReLU layers. We leave this for future work.
>
> -L249: We apologize for not defining the abbreviation. CRM stands for Certified Radius Maximization to refer to our method.
>
> -L251: For our updated experiments on CIFAR10 and Tiny-Imagenet, we are now using the same choices for h and obtaining similar results. For reproducibility, we will release the code upon acceptance.
>
> -L255: As shown by Fig. 4 in the appended document, our Lipschitz calculation algorithm scales linearly with network size thanks to the highly specialized implementation.
>
> About Footnote 4: at the time of submission, achieving the best hyper-parameter tuning was not possible due to the long run times on Tiny-Imagenet. Nevertheless, even with the suboptimal tuning of the hyper-parameter,s we are obtaining a similar performance (Table 1).

---

> > ### Comment · Reviewer_U11m · 2023-08-18
> >
> > Thank you for addressing my query. I stand by my score and recommend accepting the paper.

---

### Author Rebuttal · Authors · 2023-08-10

We would like to sincerely thank all the reviewers for their useful comments and feedback. We have improved the paper in terms of presentation, related work, notation, and technical aspects. Below, we report a list of global responses that would be useful for all the reviewers.

**A new regularization term:** To capture the inherent trade-off between natural accuracy and adversarial robustness [23], we use the cross-entropy loss for clean accuracy and use a new regularizer for adversarial robustness. Our regularizer, inspired from and built upon prior works [17,31,18,24], has the following novel features:

- It is interpretable and explicit: since our regularizer is precisely a lower bound on the distance to the decision boundary, it directly increases the input margins instead of relying on their indirect approximations, enabling a more controllable and efficient enhancement of robustness.

- Our regularizer is differentiable, and computationally efficient thanks to an efficient method to estimate the Lipschitz constant of neural networks.

- Our regularizer is dynamic, in that it uses a barrier function (the function $h$) to prioritize data points that are closer to the decision boundary, as opposed to assigning a fixed robustness margin to all data points.

**A new algorithm for Lipschitz estimation:** We develop a truly scalable and differentiable method for bounding the Lipschitz constant of neural networks. This bound is provably better than the “naive” bound (product of norms of weights) at the expense of an increase in the number of matrix norms we need to compute. To offset this cost, we develop a highly parallelized GPU-enabled implementation that reduces the practical complexity to that of the naive method (please see Figure 4 in the attached pdf). While this algorithm can be of independent interest in several other domains, we tailor it toward robust training in this paper. The ultimate bottleneck in scalability is due to the use of pairwise Lipschitz constants and not the Lipschitz constant estimation itself. We can mitigate this by using the per-class Lipschitz constant or the Lipschitz constant of the whole networks, as also done in [18,24]. We also note in our algorithm, a lot of operations for computing pair-wise Lip. or per-class Lip would remain the same and only the very last iteration would change.


**Details on Loop Transformation:** We apologize for not explaining this concept in more detail, which is the core idea behind the efficiency of the proposed Lipschitz estimation method. Historically, loop transformation has been used in the context of robust control. In our context, loop transformation refers to the operation outlined in equation after line #651, where we subtract a linear term from the nonlinear term $G\phi(Wx)$ and add it to the linear term $Hx$, resulting in an equivalent representation of the layer $h(x)$:

$h(x) = (H + \frac{\alpha+\beta}{2}GW)x + G\psi(W x)$

Where now $\psi$ is the “transformed” nonlinearity. Applying the naive bound on this equivalent representation will provably improve the Lipschitz bound. In the appendix, we have provided both theoretical and intuitive justification as to why this “loop transformation” improves the naive bound. In section A.3.1, we have also proven the optimality of this transformation in the sense that, if we shift the nonlinearity by $\gamma x$, where $\gamma$ is a scalar, the optimal value of $\gamma$ that results in the smallest upper bound is $\gamma=(\alpha+\beta)/2)$. Intuitively, we are shifting the nonlinearities to make them slope-restricted in a symmetric interval, which would minimize the conservatism of the naive bound (please see lines #205-#209)

The term loop transformation comes from a control-block-diagram interpretation of the above technique. For detailed descriptions of loop transformations, please see the book by Desoer and Vidyasagar [R1, p50]. We will cite this reference in the revised manuscript.

[R1]:  Desoer, Charles A., and Mathukumalli Vidyasagar. Feedback systems: input-output properties. Society for Industrial and Applied Mathematics, 2009.

**Complexity Analysis of Lipschitz Estimation Algorithm:**  We apologize for not being clear in the analysis of complexity. We have added a new section to the paper to address this. We state the final result here:

As we are utilizing the power method to calculate the norm of matrices, the main computational complexity is introduced by passing a vector/tensor through a linear layer. Thus to simplify exposition, we present the result in terms of the number of passes through the different weights of the network (as pointed out in Algorithms (1) and (2), the power method involved passing a vector though $W_i$ and $W_i^T$): For the fully connected architecture stated in equation (35), which is the architecture used for the experiments of the paper, if we had L layers and performed N iterations of the power method, the naive Lipschitz estimation bound requires $O(NL)$ passes through different weights. For our method, the number of passes is $O(NL^3)$. However, we can use the recurrent structure of the calculations to our advantage based on the explanation of Algorithm 2. By performing $O(L^3)$ concatenations in the batch dimension, we would then need to only perform $O(NL)$ passes through different weights. Figure 4 in the attached pdf confirms this: as we increase the number of layers, the Lip. calculation runtime increases almost linearly, similar to the naive bound.

---

### Decision · Program_Chairs · 2023-09-21

**Decision:**

Accept (poster)

**Comment:**

The paper introduces a novel loss function to enhance the robustness of neural networks by enlarging the input space margin. One contribution is the proposal of an efficient method for estimating the Lipschitz constant, which is simple and performs well. Experiments on standard datasets like MNIST, CIFAR-10, and TinyImageNet showcase improved certified accuracy and robust radius, establishing the method's superiority over state-of-the-art approaches. Overall, the paper is well-written, theoretically grounded, and provides valuable insights that advance the field. I recommend acceptance.